# Development of a broad-spectrum epitope-based vaccine against *Streptococcus pneumoniae*

**Md. Nahian**[1], **Muhammad Shahab**[2], **Md. Rasel Khan**[1], **Shopnil Akash**[3], **Tanjina Akhtar Banu**[4], **Murshed Hasan Sarkar**[4], **Barna Goswami**[4], **Sanjana Fatema Chowdhury**[4], **Mohammad Ariful Islam**[1], **Ahmed Abu Rus'd**[1], **Shamima Begum**[1], **Ahashan Habib**[4], **Aftab Ali Shaikh**[4], **Jonas Ivan Nobre Oliveira**[5], **Shahina Akter**[4]*

1 Department of Microbiology, Jagannath University, Dhaka, Bangladesh, 2 State Key Laboratories of Chemical Resources Engineering, Beijing University of Chemical Technology, Beijing, China, 3 Computational Biology Research Laboratory, Department of Pharmacy, Daffodil International University, Dhaka, Bangladesh, 4 Bangladesh Council of Scientific and Industrial Research (BCSIR), Dhaka, Bangladesh, 5 Departamento de Biofísica e Farmacologia, Universidade Federal do Rio Grande do Norte, Natal, RN, Brazil

* shupty2010@gmail.com

**Data Availability Statement:** The primary data used in this study was retrieved from NCBI, with the following accession numbers: Choline binding protein A (Accession: CJD51254.1) and

## Abstract

*Streptococcus pneumoniae* (SPN) is a significant pathogen causing pneumonia and meningitis, particularly in vulnerable populations like children and the elderly. Available pneumonia vaccines have limitations since they only cover particular serotypes and have high production costs. The emergence of antibiotic-resistant SPN strains further underscores the need for a new, cost-effective, broad-spectrum vaccine. Two potential vaccine candidates, CbpA and PspA, were identified, and their B-cell, CTL, and HTL epitopes were predicted and connected with suitable linkers, adjivant and PADRE sequence. The vaccine construct was found to be antigenic, non-toxic, non-allergenic, and soluble. The three-dimensional structure of the vaccine candidate was built and validated. Docking analysis of the vaccine candidate by ClusPro demonstrated robust and stable binding interactions between the MEV and toll-like receptor 4 in both humans and animals. The iMOD server and Amber v.22 tool has verified the stability of the docking complexes. GenScript server confirmed the high efficiency of cloning for the construct and *in-silico* cloning into the pET28a (+) vector using SnapGene, demonstrating successful translation of the epitope region. Immunological responses were shown to be enhanced by the C-IMMSIM server. This study introduced a strong peptide vaccine candidate that has the potential to contribute to the development of a rapid and cost-effective solution for combating SPN. However, experimental verification is necessary to evaluate the vaccine's effectiveness.

## Introduction

*Streptococcus pneumoniae*, usually called pneumococcus, is a Gram-positive bacterium. It resides in the throat and can be in the nasopharynx of healthy individuals and transferred

Pneumococcal surface protein A (Accession: VMA87220.1). All relevant data are included within the manuscript and the supporting information files. For any inquiries regarding the data, please contact the designated non-author data access point: Lincon Mazumder, Graduate Student, Department of Biology, Texas A&M University, College Station, USA (email: lincon@tamu.edu).

**Funding:** The author(s) received no specific funding for this work.

**Competing interests:** The authors have declared that no competing interests exist.

through respiratory droplets from carriers [1]. It continues to be a significant contributor to life-threatening illnesses, including pneumonia and meningitis. Additionally, it can cause further infections by following respiratory diseases caused by viruses like COVID-19 [2]. Pneumonia kills 1 child under 5 years old every 43 seconds. It is the primary reason for child death under 5 years old than any of the diseases in the current world. It kills over 700000 children under five years old, which means almost 2000 per day in 2021 [3]. In 1990, pneumonia killed 600000 people aged over 70+, whereas in 2019, this disease killed 1.23 million people over 70 + age [4]. Beyond the immediate threat of pneumonia, SPN can lead to a cascade of severe health complications, including bronchitis, brain abscess, ear infections, blood poisoning, bone infections, skin infections, endocarditis, myocarditis, ocular infections, inflammation of the abdominal lining, and acute sinus infections [5–8]. In 2018, the "European Centre for Disease Prevention and Control" reported about 24,663 aggressive pneumococcal disease (IPD) incidents across the European Union and European Economic Area. Meanwhile, in the United States, the "Centers for Disease Control and Prevention" (CDC) reported that the rate of incidence was 7 cases among 100,000 people nationwide [9]. "The World Health Organization" (WHO) revealed that pneumococcal infections resulted in 1.6 million fatalities globally in 2005, with a significant number of these deaths occurring among children below 2 years old and adults 65 years old and above [10]. Furthermore, in 2008, South Asia alone experienced 113,000 deaths due to pneumococcal pneumonia, including the disease meningitis [11].

The uprise level of antibiotic-resistant pneumococci has become a significant concern globally. In the United States, research conducted in 2008 revealed that pneumococci had already developed obstruction to a wide range of antibiotics, including penicillin, quinolones, macrolides, and cephalosporins [12]. Research in China has identified alarmingly high resistance rates in *S. pneumoniae* to commonly used antibiotics. The resistance rates were particularly concerning for clindamycin (95.8%), erythromycin (95.2%), tetracycline (93.6%), and trimethoprim/sulfamethoxazole (66.7%). These findings highlight the critical need for stricter antibiotic use practices and ongoing research efforts to tackle the growing problem of bacterial resistance [13]. Studies on antibiotic resistance reveal a substantial healthcare burden. It's estimated to have resulted in an additional 32,398 outdoor patient presence, including 19,336 hospitalizations. This translates to a significant increase in direct medical costs [14]. This significant expense associated with antibiotic treatment, including the escalating resistance of pneumococcus to existing antibiotics, underscores the importance of vaccination required for *S. pneumoniae* as the most effective approach for preventing disease [15].

Currently, available pneumococcal vaccines fall into two categories. However, both types target a restricted amount of capsular polysaccharides located on the bacteria's surface, limiting their overall effectiveness. The 23-valent pneumococcal polysaccharide vaccine (PPV) includes 23 distinct capsular polysaccharides unique to different *S. pneumoniae* serotypes [16], while the 7-, 10-, or 13-valent pneumococcal conjugate vaccine (PCV) incorporates these prevalent polysaccharides linked to a transport protein [16, 17]. The PPV offers a broad range of coverage but doesn't safeguard vulnerable populations below two years of age. In contrast, PCV triggers robust protective immunity for newborn children but offers a narrower range of coverage and comes with a hefty production expense, as well as the need for multiple injections [18, 19]. However, the effectiveness of current vaccines is limited by the emergence of serotypes that are not covered by vaccines, which can replace vaccine-targeted serotypes [20, 21]. This underscores the need for a novel, economically efficient vaccine that provides more comprehensive immunity in contrast to the more diverse variety of *S. pneumoniae* strains [21, 22].

A promising method for vaccinating against *Streptococcus pneumoniae* involves an epitope-based strategy. This approach utilizes computational techniques that are both time- and cost-effective, allowing for the targeted selection of potential epitopes from specific proteins [23].

Unlike traditional vaccines designed for specific serotypes, epitope-based vaccines offer a broader protective range [24]. The popularity of this technique in vaccine construction is rising due to advancements in next-generation sequencing including the biological databases availability [25]. By identifying optimal ligands for human leukocyte antigens, this method precisely determines T-cell epitopes essential for vaccine development [26]. Driven by this potential, researchers have extensively studied various pneumococcal proteins over the years, aiming to develop a vaccine that effectively combats pneumococcal diseases [27]. These studies have included surface proteins such as Choline-Binding Protein A (CbpA), Pneumococcal surface protein A (PspA), Pneumococcal Histidine Triad (Pht) protein family (PhtA, PhtB, PhtC, PhtD), Pneumolysin (PLY), and ABC transporters (PsaA, PiuA, and PiaA) [27–29]. Recent research has focused on various pneumococcal proteins with roles in colonization and virulence, and these proteins have shown the ability to stimulate protective immune responses in both animal models and humans [30–32].

Through a comprehensive literature review and bioinformatic screening, our study identified CbpA and PspA as promising candidates for epitope-based vaccine design. CbpA, also known as PspC, is a major adhesin in *S. pneumoniae*, facilitating colonization and disease progression. This protein's ability to interact with the polymeric immunoglobulin receptor (pIgR) enables it to cross the epithelial cells in the human nasopharynx, a key step in the development of meningitis [33]. Moreover, CbpA's interaction with the host protein factor H reduces the activity of the other possible complement pathway, contributing to immune evasion [1]. The antigenic variation observed in CbpA suggests its role in immune evasion, making it a challenging but crucial target for vaccine design [34]. CbpA can stimulate CD4+ T-lymphocyte-dependent antibody production when used as an immunogen, offering significant protection [35]. PspA, on the other hand, exhibits high conservation throughout different strains of pneumococcus and stimulates a protective immune response, making it a valuable component of vaccine formulations [36]. It contributes to virulence through adhesion, biofilm formation, and upper respiratory tract colonization, indicating a specific role in invasive disease [37]. Research on PspA as a vaccine target spans over three decades. A phase I clinical study showed that a genetically engineered family 1 PspA protein, generated in *Escherichia coli*, induced an immune response when administered to humans. Additionally, the study demonstrated passive protection in mice against challenges with pneumococcal serotypes 3, 6A, and 6B [38].

A crucial factor in vaccine development is the incorporation of adjuvants. These substances improve the immunological response, making it faster, stronger, and longer-lasting [39]. Recent research indicates that melittin, an essential part of bee venom, may be a useful addition to vaccine development, especially for multi-epitope vaccines [40]. Evidence has demonstrated that Melittin augments the Th1 cell response of the immune system, a critical factor in combating infections [41]. Furthermore, it has the ability to modulate the synthesis of cytokines, therefore diminishing regulatory T cells and augmenting pro-inflammatory components. The maintenance of this equilibrium is crucial for a robust immunological response [42, 43]. Furthermore, the nasal administration of melittin has shown efficacy, which may result in a more prolonged immunological response in comparison to the use of the antigen alone [44, 45]. It works by releasing molecules known as damage-associated molecular patterns (DAMPs), which in turn activate particular toll-like receptors (TLRs) and set off an immunological reaction. This activation signifies cellular damage, prompting the immune system to respond [46].

This research focuses on developing a multi-epitope vaccine that effectively triggers the innate and adaptive immune systems within the host, offering optimal protection against *S. pneumoniae* (SPN). Therefore, the reference sequence of CbpA and PspA protein was chosen according to their predicted antigenic score and subcellular localization. These proteins were

then analyzed to identify potential epitopes for cytotoxic B cells, T cells (CTLs), helper T cells (HTLs),. We then assembled the vaccine by strategically combining the most promising CTL, HTL, and B-cell epitopes with adjuvant, PADRE sequence, and appropriate linkers. To predict the vaccine's potential to activate the immune system, we modeled its structure and virtually docked it with TLR4. We further evaluated the docked complex using computer simulations to assess its stability and functionality. In addition to structural assessment, immune simulations were performed to determine the vaccine's ability to generate a lasting immunological memory against *S. pneumoniae* infections. Finally, we performed *in silico* cloning to evaluate the vaccine's potential for successful expression and translation. One key advantage of our proposed vaccine compared to existing peptide vaccines is that the selected epitopes are predicted to induce immune responses in both humans and mice. This feature could significantly reduce the costs associated with *in vivo* testing required to validate the vaccine's efficacy.

## Methodology

Fig 1 describes the sequential methodology used to develop the multi-epitope vaccine targeting *S. pneumoniae*.

### Identifying protein candidates for vaccine development

To identify potential vaccine targets for SPN, a comprehensive literature review encompassing both experimental and computational studies was conducted. Given the vast and complex genome of *Streptococcus pneumoniae* and the fact that not all proteins trigger a significant immune response in the humans, the focus was on previously characterized proteins with known or potential immunogenic properties [47–49]. This initial step allowed for narrowing the focus to proteins that were already characterized as promising candidates for vaccination due to their capacity to provoke an immune response (S1 Table).

### Bioinformatic screening and candidate protein selection

After compiling a list of potential virulent proteins from the literature, a bioinformatic screening process was conducted to further refine the selection. To assess the antigenic qualities of these proteins, VaxiJen version 2.0 (http://www.ddg-pharmfac.net/vaxijen/VaxiJen/VaxiJen.html) was used [50]. The VaxiJen tool is commonly used for the prediction of protein antigenicity, which is crucial for determining their ability to induce an immune response [50]. Only proteins with a high antigenic score were considered for further analysis. In addition to antigenicity, the subcellular localization of the proteins was assessed using PSORTb (https://www.psort.org/psortb/) [51]. Subcellular localization is a key factor in vaccine development, as surface-exposed proteins are more readily recognized by the immune system and are more likely to serve as effective vaccine targets. Proteins anticipated to be extracellular or linked to the bacterial surface were prioritized due to their increased likelihood of being recognized by the host immune system [51]. This combined analysis resulted in the selection of top-scoring candidate proteins with the highest predicted antigenicity and confirmed extracellular localization, making them ideal targets for further vaccine development.

### Discovery of B cell specific epitopes

Linear B-cell antigenic determinants were determined using several servers, including LBtope (http://crdd.osdd.net/raghava/lbtope/) [52], ABCpred (http://crdd.osdd.net/raghava/abcpred/) [53], the Emini surface accessibility prediction tool (http://tools.iedb.org/bcell/result), and Ellipro (http://tools.iedb.org/ellipro/) from the server Immune Epitope Database (IEDB) [54].

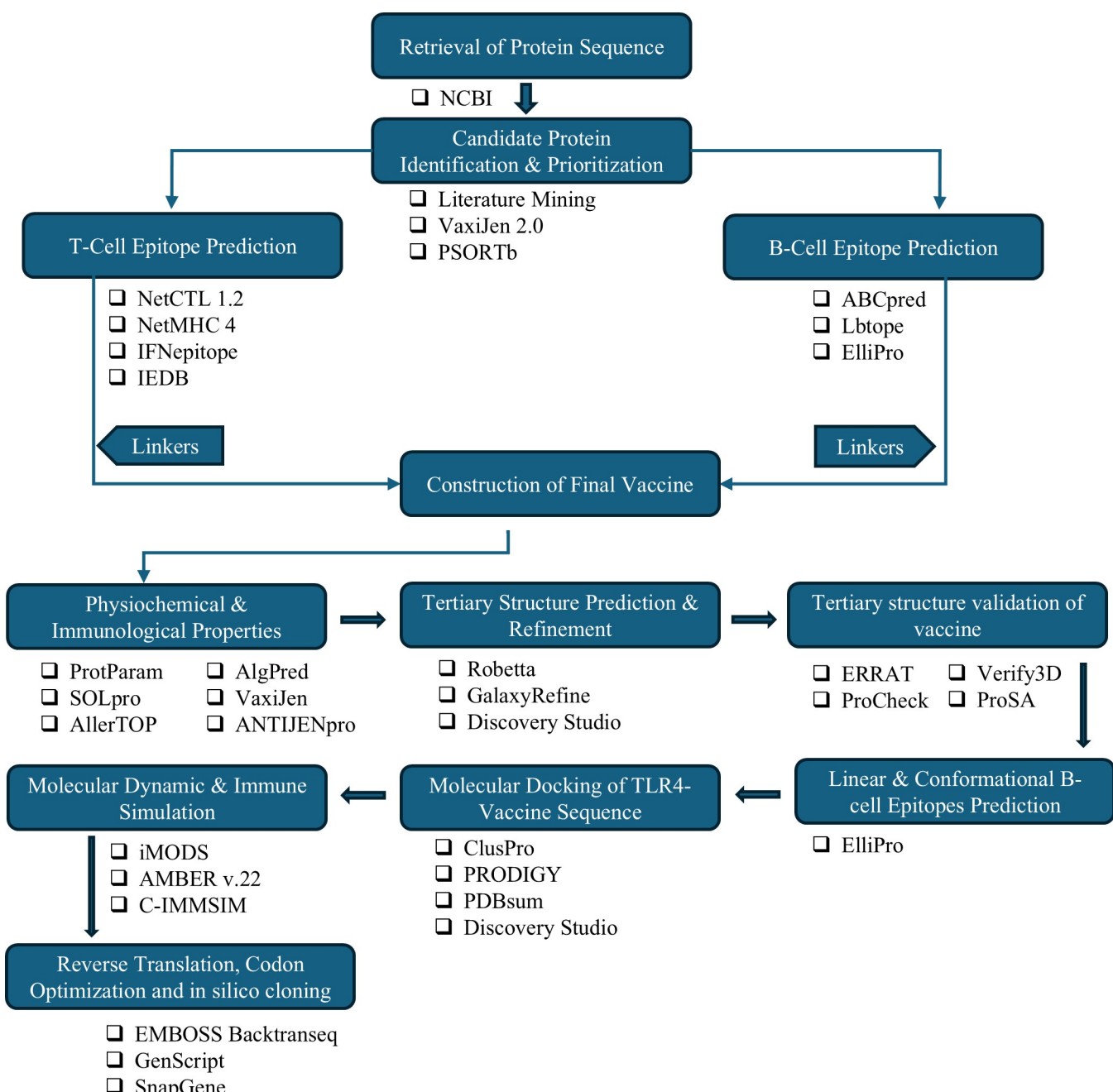

**Fig 1. Multi-epitope vaccine development process against *streptococcus pneumoniae*.** This flowchart outlines the key steps in developing a multi-epitope vaccine targeting *S. pneumoniae*, highlighting the tools and servers used at each stage.

The LBtope server, which employs support vector machine (SVM) technology, achieves a predictive accuracy of approximately 81% [52]. ABCpred, a neural network-based program, predicts linear epitopes where rate of accuracy is almost 65.93% and scores near one indicate a greater probability to be an epitope [53]. Epitopes which are surface accessible was predicted by using the Emini method with a cutoff value of 1.0.

## Detection and analysis of MHC class I epitopes

The NetCTL 1.2 server (https://services.healthtech.dtu.dk/services/NetCTL-1.2/) was used to screen the chosen proteins with the default parameters in order to anticipate 9-mer CTL epitopes that target 12 human HLA Class-I superfamily alleles (A1, A2, A3, A24, A26, B7, B8, B27, B39, B44, B58, and B62) [54]. The epitope length was set to 9-mers, consistent with the trained dataset of the NetCTL 1.2 server, which utilized 886 known 9-mer MHC I ligands and achieved an average AUC score of 0.941[55]. Epitopes demonstrating a human supertypes binding affinity with higher value were subjected to further screening using the following methods: 1) TMHMM server for predicting transmembrane location [56], 2) VaxiJen v2.0 server for scoring antigenicity [50], 3) AllerTOP v.2.0 web server for forecasting the likelihood of allergy [57], and 4) ToxinPred web server for forecasting the likelihood of being toxic [59]. Using the NetMHC 4 server (https://services.healthtech.dtu.dk/services/NetMHC-4.0/), the binding capacity of the epitopes to mouse alleles (H-2-Db, H-2-Dd, H-2-Kb, H-2-Kd, H-2-Kk, and H-2-Ld) that had passed the screening and satisfied the requirements of being topologically external, immunogenic, allergy free, and not toxic was then prioritized [58]. Epitopes that exhibited good findings were further verified by the help of IEDB class I immunogenicity server to assess their binding to human HLA class I. Ultimately, the epitopes with the broadest HLA coverage were selected as the final candidates.

## Detection and investigation of MHC class II epitopes

The IEDB MHC II binding server (http://tools.iedb.org/mhcii/) was employed to forecast epitopes derived from the chosen antigenic proteins [59]. Initially, 15-mer HTL epitopes were identified for mouse alleles, selecting those with a percentile rank below 10. Subsequently, the epitopes were filtered using the TMHMM, AllerTOP v.2.0, VaxiJen v2.0, and ToxinPred servers, as done during the initial CTL screening [50, 56, 57, 60]. The next step involved screening the epitopes using the IFNepitope web server (http://crdd.osdd.net/raghava/ifnepitope) for identifying those capable of generating IFN-γ production [61]. To ensure human compatibility, epitopes with promising results underwent further validation using IEDB class II immunogenicity tool for binding to human HLA class II molecules. Similar to the CTL screening process, epitopes with the broadest coverage of HLA class II alleles were chosen as the desired HTL epitopes, also subsequently analyzed for population coverage.

## Docking of the epitopes

The binding affinity and interaction between cytotoxic T lymphocyte (CTL) and helper T lymphocyte (HTL) epitopes and their corresponding major histocompatibility complex (MHC) alleles were computationally evaluated using molecular docking. This analysis was conducted to assess the structural compatibility of these molecules and to forecast the possible immunological response that the epitopes may trigger [62]. To perform these docking simulations, three-dimensional (3D) structures of both the peptide epitopes and MHC alleles were required. The 3D structures of high-scoring epitopes were predicted using the PEP-FOLD (https://bioserv.rpbs.univ-paris-diderot.fr/services/PEP-FOLD/) server [63], while the structures of MHC alleles were obtained from the RCSB Protein Data Bank (https://www.rcsb.org/) [64]. Molecular docking simulations were executed using the Cluspro server (https://cluspro.bu.edu/publications.php) [65], and the resulting docked complexes were then visualized and refined using BIOVIA Discovery Studio Visualizer. Lastly, the PRODIGY (https://rascar.science.uu.nl/prodigy/) server was used to determine the binding affinity between the epitopes and MHC alleles [66, 67].

## Population coverage and epitope conservancy analysis

Significant differences are found between expression and distribution of HLA alleles exist among populations of various ancestries. This study evaluated the degree of conservation of MHC-I and MHC-II molecules across different viral variants and the population coverage by these molecules using the IEDB Analysis Resource [59]. The percentage of the number of individuals living in a specific area covered by the selected epitopes was estimated employing the IEDB Population Coverage Tool (http://tools.iedb.org/population/) [59].

## Vaccine formulation and structural insights

The vaccine design commenced with Melittin, a candidate adjuvant, fused to the N-terminus via an EAAAK linker, chosen for its ability to enhance both solubility and adjuvant activity [68]. Subsequently, a PADRE sequence was incorporated to provide a robust T-helper epitope [69]. KK linkers were used to connect the B-cell epitopes, which serve as target sites for cathepsin B, an essential enzyme in antigen processing for MHC-II presentation [70]. By connecting epitopes with KK linkers, individual peptide components can be specifically targeted to antibodies, preventing the formation of antibodies against the linker itself [71]. GPGPG linkers were used for connecting HTL epitopes; the glycine-rich composition of this linker enhances the solubility of the resultant molecule while facilitating independent movement and ensuring high accessibility and flexibility for neighboring domains [72]. GGGGS linkers were employed to connect CTL epitopes, as their flexibility and hydrophilicity are crucial for efficient antigen processing and presentation [73]. A final PADRE sequence was included at the C-terminus to further amplify the immune response.

## Examination of the vaccine sequence properties

After the multi-epitope-based vaccine was designed and constructed by carefully combining certain amino acid sequences with appropriate linkers, a comprehensive evaluation was conducted to analyze the construct's physicochemical and immunological properties. This analysis utilized several bioinformatics tools to examine key features. Physicochemical parameters essential for verifying the vaccine design, including molecular weight, isoelectric point (pI), half-life, instability index, aliphatic index, and GRAVY, were computed using the ExPASy ProtParam tool (http://web.expasy.org/protparam/) [74]. The tool calculated the molecular weight, which is the total of the atomic weights of all amino acids in the protein. This result offers valuable information on the size of the vaccine construct, which is crucial for assessing its biological activity and delivery capability. This isoelectric point (pI) of the proteins is the pH where they have no net charge. The isoelectric point affects the solubility and precipitation behavior of the vaccine, influencing its formulation and storage stability. The predicted half-life offers insights into the vaccine's stability and longevity in different biological environments, which is essential for guaranteeing a sufficient duration of the immunological response. The instability index measures the protein's *in vitro* stability; a result of 40 or less indicates a stable vaccine candidate, essential for maintaining structural integrity during production and storage. The thermal stability of the vaccine is determined by the aliphatic index, which quantifies the proportional volume of aliphatic side chains. This coefficient guarantees the vaccine's ability to withstand temperature fluctuations during handling and transportation. The grand average of hydropathicity (GRAVY) assesses the balance between hydrophobic and hydrophilic regions in the protein, affecting its solubility and interaction with biological membranes, thus impacting the vaccine's efficacy and bioavailability. Additionally, the solubility of the vaccine construct upon overexpression in *E. coli* was evaluated using the SOLpro server (https://

scratch.proteomics.ics.uci.edu/), indicating the protein's potential for high solubility, which is critical for efficient production and purification processes [75].

To evaluate antigenicity, two software programs were utilized: VaxiJen v2.0 (http://www. ddg-pharmfac.net/vaxijen/VaxiJen.html) along with ANTIGENpro (http://scratch. proteomics.ics.uci.edu/) [50, 76]. VaxiJen offers antigenicity prediction without depending on sequence alignments. It uses a primary threshold of 0.4 and achieves an accuracy range of 70% to 89%, which can vary depending on the targeted organism [50]. Similarly, the SCRATCH server's ANTIGENpro is also an alignment-free, pathogen-independent protein antigenicity prediction with a 76% accuracy rate and a 0.5 threshold [76]. To assess the probable allergenicity of identified vaccine construct, two *in silico* tools, AlgPred (https://www.imtech.res.in/ raghava/algpred/) [77] as well as AllerTOP (http://www.ddg-pharmfac.net/AllerTOP/) [63], were applied. AlgPred employs a multifaceted approach, including Support Vector Machines (SVM), motif identification, and BLAST sequence similarity searches, along with hybrid methods, to predict allergenicity and pinpoint IgE epitopes (potential allergy-causing regions) [77]. AllerTOP, on the other hand, operates independently of sequence alignments and leverages protein physicochemical properties for allergen prediction [57].

## Modeling, refining and validating the three-dimensional structure of vaccine

For predicting the three-dimensional (3D) structure of the designed vaccine, Robetta server (http://robetta.bakerlab.org/) was employed [78]. This server leverages either comparative modeling, which utilizes known protein structures for reference, or ab initio approaches, which build structures from scratch [78]. Subsequently, the GalaxyRefine server was utilized for further optimization of the most promising predicted structure [79]. The standards of the improved model was then thoroughly reviewed using a suite of bioinformatics tools which includes, ERRAT, PROCHECK, and VERIFY-3D available on the SAVES server (https://saves. mbi.ucla.edu/) [80], plus ProSA server (https://prosa.services.came.sbg.ac.at/prosa.php) [81]. VERIFY-3D analyzes the compatibility between a protein's 3D structure. It achieves this by comparing the model's structure to a database of known protein profiles [82]. ProSA (Protein Structure Analysis), on the other hand, offers a more holistic evaluation of the model's quality. This tool assesses whether the predicted structure exhibits the characteristics typically observed in naturally occurring proteins [81].

## Disulfide modification for improved vaccine peptide structural integrity

Disulfide linkages increase the free energy of the denatured state and decrease conformational entropy, which improves protein stability. The incorporation of novel disulfide bonds has been acknowledged as a significant biotechnological approach to augment the thermal stability of inherently folded proteins. The Disulfide by Design 2 server (http://cptweb.cpt.wayne.edu/ DbD2/) was employed to enhance the structural integrity of the intended vaccination by including disulfide linkages [83]. Upon substitution of each amino acid residue with cysteine, this web server can identify the pair of residues that are capable of forming a disulfide bond. A set of criteria was employed to determine possible combinations of residues (2.2 energy value and -87 to +97 chi3 value), which were subsequently modified by the addition of a cysteine residue.

## Prediction of both linear & discontinuous B-cell epitopes

For stimulating a powerful humoral immune response, a vaccine construct must possess effective B-cell epitopes within its protein domains. These epitopes has the possibility to be the

linear stretches of amino acids or discontinuous structures formed by residues brought together in the 3D conformation [84]. ElliPro (http://tools.iedb.org/ellipro/), a well-regarded tool with a high AUC value of 0.732, was employed to identify such epitopes within the optimized selected model of multi-epitope vaccine. The analysis utilized ElliPro's default parameters [85].

## Interaction studies through molecular docking and dynamic simulations with TLRs

For checking the probable interaction among the designed multi-epitope-based vaccine and toll-like receptors (TLRs), protein-protein docking simulations were completed by ClusPro web server (https://cluspro.org/login.php) [65]. The refined vaccine model acted as the ligand for docking with crystallized structures of human TLR-4 (PDB ID: 4G8A) as well as mouse TLR-4 (PDB ID: 2Z64), downloaded from the Protein Data Bank (PDB) [64]. Following docking analysis, complexes with the most favorable energy scores were selected and downloaded for further investigation. The docked molecules were visualized with Discovery Studio Visualizer software. To estimate the binding strength and visualize the standard key residues between the vaccine and TLRs, the PRODIGY server (https://nestor.science.uu.nl/prodigy) was employed [67]. Finally, the complexes exhibiting the strongest and powerful binding affinities were tested to stability and dynamic analyses using the iMod web server (http://imods.Chaconlab.org/), and Amber v.22 tool (http://imods.Chaconlab.org/) [86, 87]. This analysis assessed the physical movements within the docked receptor-vaccine complexes.

## Simulation of immune response

The predicted immunological response of the developed vaccine candidates was assessed utilizing the C-IMMSIM webserver (https://kraken.iac.rm.cnr.it/C-IMMSIM/index.php) [88]. The PDB files for the proposed vaccine model were put on to the server for this experiment. Predefined simulation parameters were employed, including a randomly selected seed of 12,345, a simulation volume of 50, and a total of 1100 simulation steps. Three virtual injections were simulated at specific time points (1, 84, and 168), with each step representing an eight-hour period. Notably, no lipopolysaccharide (LPS) was used, and the shortest interval of 30 days was maintained between two injections.

## Vaccine virtual expression & cloning

Initially, the amino acid sequence was transformed into a nucleotide sequence using the EMBOSS Backtranseq web server (https://www.ebi.ac.uk/Tools/st/emboss_backtranseq/) [89]. Subsequently, the codons used in the final vaccine design were modified to ensure effective expression in human cells. Optimization was conducted using the GenSmart Codon Optimization Tool server (https://www.genscript.com/tools/gensmart-codon-optimization) [90]. Notably, the procedure guaranteed the prevention of cleavage sites for two restriction enzymes, NdeI and XhoI, in order to enable their production in the *E. coli* system. The quality of the optimised codons was evaluated using the rare codon analysis server (https://www.genscript.com/tools/rare-codon-analysis) [91]. The optimized sequence was integrated into a plasmid vector, pET-28a (+), using *in silico* cloning using SnapGene software [92]. To simplify this process, specific restriction enzyme recognition sites (NdeI and XhoI) were constructed at the N-terminus and C-terminus of the sequence.

## Results

### Antigenic profiling and subcellular localization of target protein

Initially, 23 proteins with potential vaccine properties against *Streptococcus pneumoniae* were identified through literature mining. Bioinformatic screening using VaxiJen and PSORTb was conducted for all 23 proteins to evaluate their antigenicity and subcellular localization (S1 Table). Among these, CbpA (CJD51254.1) and PspA (VMA87220.1) emerged as the most promising candidates. CbpA and PspA exhibited high antigenicity scores of 0.7721 and 0.6565, respectively, indicating their strong potential as antigens. Further analysis using the PSORTb tool confirmed that both CbpA and PspA are predicted to be extracellular, each with a localization score of 9.73. This extracellular orientation suggests that these proteins are well-positioned to interact with the host immune system, enhancing their suitability as vaccine targets. Consequently, CbpA and PspA were selected as the primary candidates for the development of a multi-epitope-based vaccine.

### Discovery of B cell epitopes

B cell epitopes were extracted using a suite of tools including ABCpred, LBTope, the Emini surface accessibility prediction tool, and the Ellipro server. The identified B-cell epitopes for Choline-binding protein A (CbpA) are detailed in (Table 1 of S2 Table), while those for Pneumococcal surface protein A (PspA) are presented in (Table 2 of S2 Table). These epitopes were carefully selected for their potential to elicit strong immune responses, thereby serving as crucial components in the development of the multi-epitope vaccine. These epitopes were checked for further evaluation for antigenicity, allergenicity, toxicity, and transmembrane region localization utilizing the VaxiJen, AllerTop, ToxinPred, and TmHMM web servers, respectively. Following this comprehensive evaluation, epitopes exhibiting the most favorable scores across all defined criteria were chosen for further vaccine development.

### Detection and analysis of MHC class I epitopes

A two-step filtering process was employed to identify candidate CTL epitopes within the CbpA and PspA proteins. The first stage utilized the NetCTL1.2 server to identify potential CTL epitopes, resulting in an initial set of 261 and 147 candidates originating from CbpA and PspA, respectively. These epitopes were then subjected to a first-stage filtration to assess antigenicity, allergenicity, non-toxicity, and the presence of transmembrane helices. Subsequently, the second-stage filtration evaluated the epitopes for optimal binding affinity with mouse alleles. From the 408 candidates identified in the first stage, 27 CTL epitopes (17 from CbpA and 10 from PspA) were selected, as detailed in (Tables 1.1 and 2.1 in S3 Table). Subsequently, 12 CTL epitopes (8 from CbpA and 4 from PspA) were finalized through the second filtration step, based on their binding affinities with mouse alleles, as outlined in (Tables 1.2 and 2.2 in S3 Table).

### Detection and analysis of MHC class II epitopes

Determination of Helper T Lymphocyte (HTL) epitopes for the chosen proteins was initially performed with the help of Immune Epitope Database (IEDB) server. The focus was on identifying epitopes that could bind to mouse MHC-II molecules. Selection prioritized epitopes with percentile ranks below 10, indicating high predicted binding affinity. The IEDB MHC-II server yielded approximately 692 HTL epitopes. These epitopes underwent a screening process modeled after the cytotoxic T lymphocyte (CTL) first-stage filtration. 63 unique HTL epitopes were identified with their desired mice alleles. Subsequent filtration using the IFNepitope server

**Table 1. Binding affinities of CTL and HTL epitopes to MHC class I and II alleles.**

| Type of T Lymphocyte | Epitope | MHC Allele | PDB ID of MHC Allele | ΔG (kcal/mol) |
|---|---|---|---|---|
| HTL | SENTPTVTSSGQDIS | HLA-DRB1*04:01 | 5JLZ | -10.2 |
| | AAKKDYETAKKKAED | HLA-DRB1*01:01 | 5V4N | -7.4 |
| | KKDYETAKKKAEDAQ | HLA-DPA1*02:01 | 7T6I | -8.3 |
| | VNTTVDGYTVNENGE | HLA-DRB1*15:01 | 5V4M | -8.9 |
| CTL | YYLEASGAM | HLA-B*35:01 | 4LNR | -10.3 |
| | YYLNANGSM | HLA-A*24:02 | 7SRK | -11.6 |

identified 51 epitopes with positive interferon-gamma (IFN-γ) responses. Among these, 27 were associated with Choline-binding protein A (CbpA) and 24 with Pneumococcal surface protein A (PspA). Detailed data on these epitopes can be found in (Tables 1 and 2 in S4 Table).

## Epitopes to HLA alleles molecular docking

Docking between top-scored epitopes and their respective HLA alleles was performed to determine effective binding. Docking using the ClusPro server revealed highly favored molecular interactions between CTL and HTL epitopes along with their respected HLA alleles (Table 1). Negative binding affinities, identified by the PRODIGY webserver, indicated thermodynamically stable bindings and predicted a strong CTL and HTL response.

## Population coverage and epitope conservancy analysis

The Immune Epitope Database (IEDB) tool was employed to analyze the breadth of population coverage offered by the designed epitopes for both MHC-I and MHC-II alleles. The analysis identified an impressive population coverage of 98.55% for the selected MHC-I epitopes. The MHC-II epitopes achieved an even greater coverage, exceeding 99.99%. As illustrated in Fig 2, a comprehensive evaluation of all epitopes collectively resulted in a near-perfect coverage of 100%. These results indicate that the selected epitopes have the materials to be broadly effective across diverse populations, with minimal variations in coverage between ethnicities.

## Vaccine formulation and structural insights

The most highly ranked B-cell, MHC-I, and MHC-II epitopes were selected for incorporation into the final design of vaccine. Fig 3 illustrates the specific peptides chosen from CbpA and PspA for this purpose. To guarantee the construct's antigenicity while minimizing allergenicity and toxicity risks, all selected epitopes were linked using GGGGS, GPGPG, and KK linkers. In order to augment the vaccine's capacity to illicit an immunological response (immunogenicity), Melittin was included as an adjuvant. Fig 3A illustrates the application of an EAAAK linker to combine the adjuvant with the PADRE sequence. Linkers KK, GPGPG, and GGGGS were then used to connect the B-cell, MHC-II (HTL), and MHC-I (CTL) epitopes. A total of 327 amino acids comprised the final vaccine design.

## Examination of vaccine sequence properties

Analysis of the designed construct using Expasy ProtParam revealed several key properties. The estimated molecular weight was 34.52 kDa, and the predicted isoelectric point (pI) was 9.10. Stability metrics included an instability index of 26.99, an aliphatic index of 54.98, and a grand average of hydropathicity of -0.905. The predicted half-life is 30 hours in mammalian reticulocytes (*in vitro*), over 20 hours in yeast (*in vivo*), and over 10 hours in *E. coli* (*in vivo*).

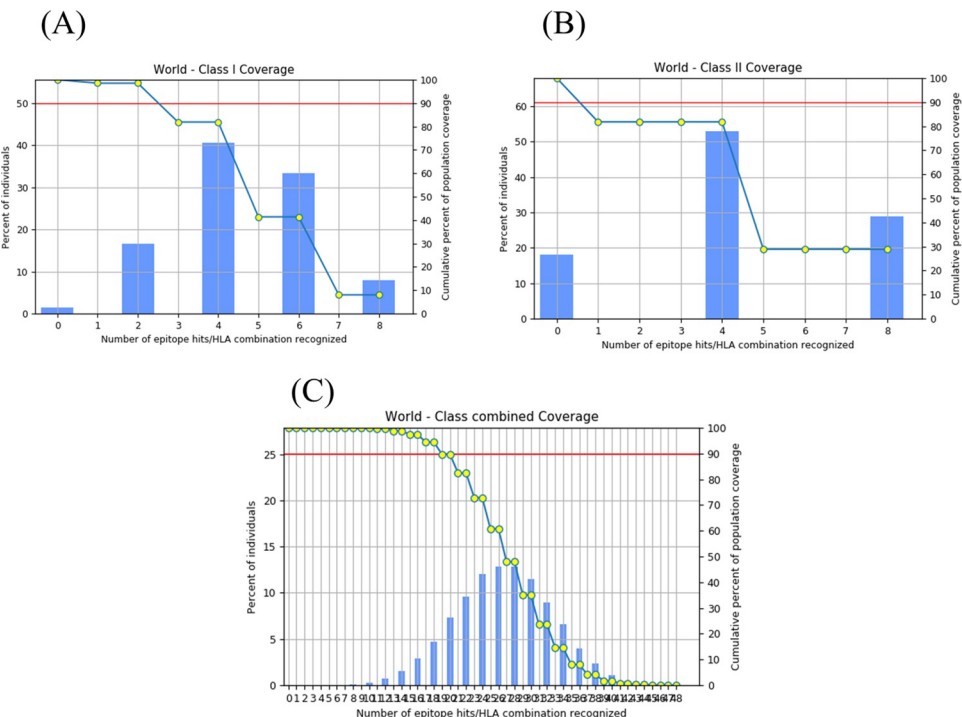

**Fig 2. Population coverage analysis of predicted epitopes.** Assessment of the population coverage achieved by the predicted epitopes for MHC-I (A), MHC-II (B), and combined MHC-I and MHC-II (C) alleles. The horizontal axis in the graph depicts the frequency of epitope hits per recognized HLA combination, while the vertical axis represents the proportion of persons and the total proportion of population coverage.

Moreover, the SOLpro server forecasted a quite high solubility (0.925991) when overexpressed in *E. coli*. To assess the construct's potential as a vaccine candidate, its antigenicity was evaluated using VaxiJen and ANTIGENpro, resulting in scores of 1.37 and 0.89, respectively, indicating good antigenic potential. Finally, analyses with AllerTOP and AlgPred servers confirmed the non-allergenic nature of the multi-epitope vaccine.

### Vaccine three-dimensional structure modeling, refining and validation

The Robetta server used a comparative modeling method to estimate the tertiary structure of the vaccine construct. The initial model was then further optimized using the GalaxyRefine server. Model 1 was selected as the optimal refined vaccine structure from the five refined vaccine model structures based on a comprehensive evaluation of structural quality parameters (Table 2). MODEL 1 demonstrated a high GDT-HA score of 0.9893, indicating strong structural similarity with the initial model, and a low RMSD of 0.283, suggesting minimal atomic deviation. Its MolProbity score of 1.425 reflects good overall structural quality. Despite a moderate clash score of 4.3, MODEL 1 maintained an excellent poor rotamers percentage of 0.4%. The percentage of Ramachandran favored residues is 96.6, higher than the initial and other refined models, indicating better backbone conformation. These combined factors highlight MODEL 1 (Fig 4A) as the optimal choice for our refined vaccine structure for its optimal combination of high similarity, minimal deviation, and overall structural quality.

In order to evaluate the overall quality of the crude and refined models, additional software programs were employed, including ERRAT, VERIFY-3D, PROCHECK from the SAVES server, and ProSA web server (Fig 4). These analyses consistently revealed improvements in

**(A)**

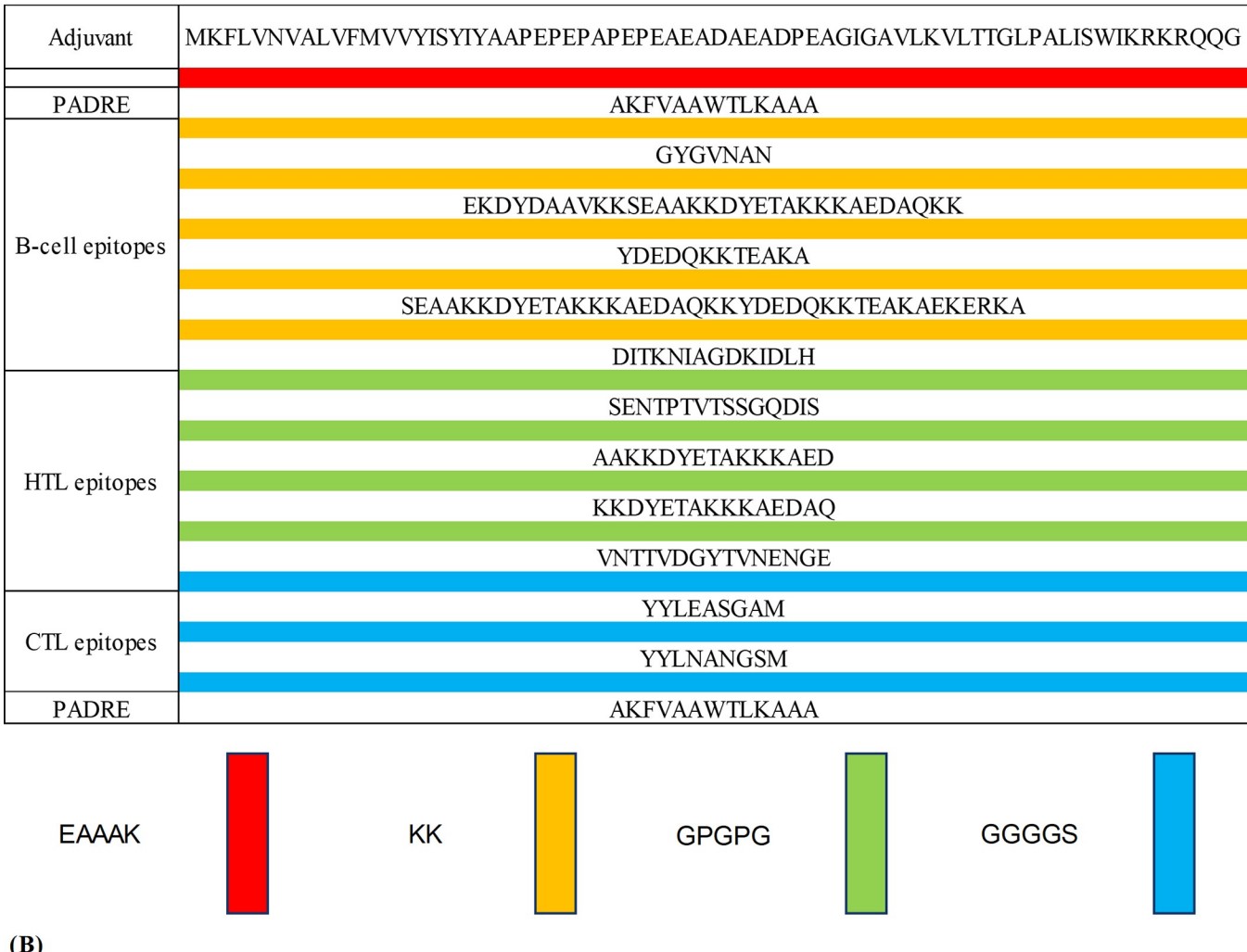

**(B)**

**MKFLVNVALVFMVVYISYIYAAPEPEPAPEPEAEADAEADPEAGIGAVLKVLTTGLPALISWIKRKRQQG EAAAKAK FVAAWTLKAAA KK GYGVNAN KK EKDYDAAVKKSEAAKKDYETAKKKAEDAQKK KK YDEDQKKTEAKA KK SEAAK KDYETAKKKAEDAQKKYDEDQKKTEAKAEKERKA KK DITKNIAGDKIDLH GPGPG SENTPTVTSSGQDIS GPGPG AAKKDYETAKKKAED GPGPG KKDYETAKKKAEDAQ GPGPG VNTTVDGYTVNENGE GGGGS YYLEASGAM GGG GS YYLNANGSM GGGGS AKFVAAWTLKAAA**

**Fig 3. Multi-epitope vaccine construct design.** (A) Schematic representation of the multi-epitope vaccine construct, illustrating the arrangement of B-cell, HTL, and CTL epitopes. (B) Sequence of the final vaccine construct, which is 327 amino acids in length. The construct incorporates various components linked together, including an adjuvant, PADRE sequence, EAAAK, GGGGS, GPGPG, and KK linkers.

various quality metrics after the refinement process. For instance, PROCHECK analysis of the Ramachandran Plot showed that in the basic model, 91.9% of residues were found in preferred regions, 5.9% in permitted regions, and 1.1% in prohibited regions. Furthermore, the improved model revealed that 93.4% of residues were in preferred regions, 5.5% in permitted regions, and 0.4% in prohibited regions ([Fig 4B]). The Z-score achieved by ProSA decreased

**Table 2. Structural parameters of refined vaccine models.**

| Model | GDT-HA | RMSD | MolProbity | Clash Score | Poor Rotamers | Rama Favored |
|---|---|---|---|---|---|---|
| Initial | 1.0000 | 0.000 | 1.364 | 2.8 | 0.4 | 95.7 |
| MODEL 1 | 0.9893 | 0.283 | 1.425 | 4.3 | 0.4 | 96.6 |
| MODEL 2 | 0.9946 | 0.262 | 1.557 | 5.3 | 0.4 | 96.0 |
| MODEL 3 | 0.9893 | 0.300 | 1.526 | 5.7 | 0.8 | 96.6 |
| MODEL 4 | 0.9817 | 0.301 | 1.439 | 4.1 | 0.0 | 96.3 |
| MODEL 5 | 0.9862 | 0.290 | 1.571 | 5.5 | 0.4 | 96.0 |

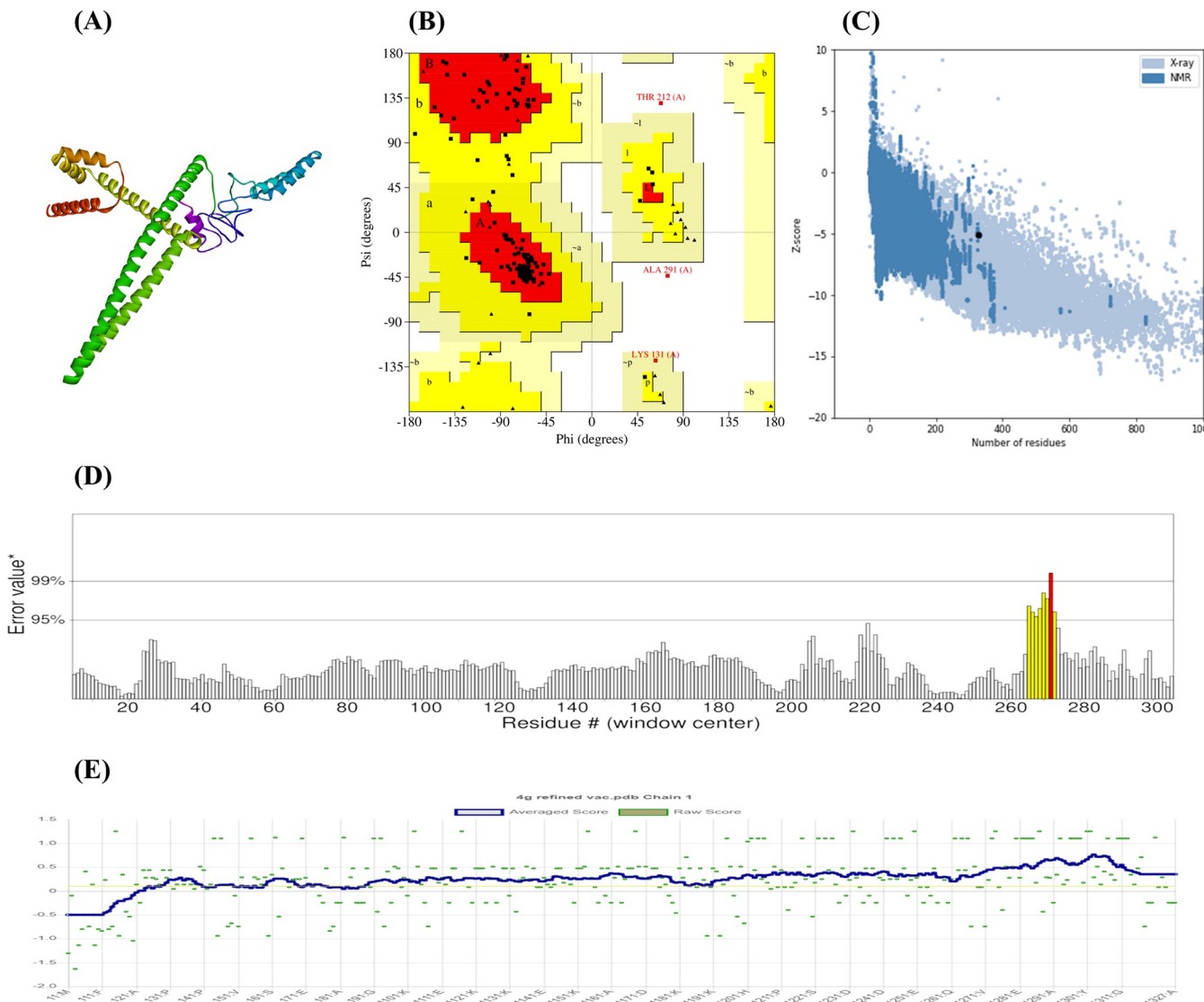

**Fig 4. Quality assessment of the refined vaccine construct.** Various assessments confirmed the high quality of the refined structure. (A) Refined 3D structure of the vaccine construct, (B) Ramachandran plot analysis, (C) Z-score distribution, (D) ERRAT quality factor, (E) VERIFY-3D score. These results indicate that the refined structure is accurate and reliable, providing a solid foundation for further analysis and simulations.

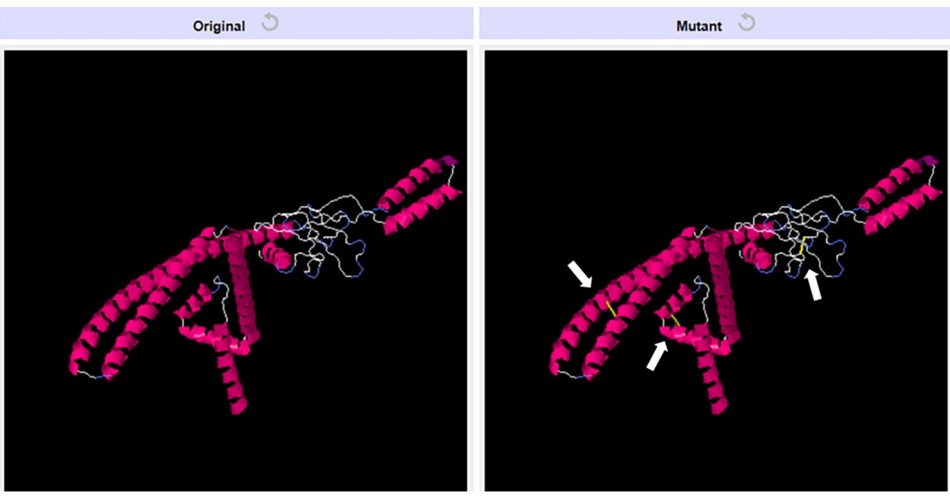

**Fig 5. Vaccine constructs with mutated structure.** The left panel shows the original vaccine construct, while the right panel displays the mutant version. The white arrows in the mutant structure highlight the locations where disulfide bonds (yellow) were introduced, leading to structural alterations.

from -5.24 in the original model to -5.12 in the improved model (Fig 4C). The original model displayed an ERRAT quality factor of 96.11, which subsequently rose to 97.46 in the updated model (Fig 4D). The results of the VERIFY-3D analysis showed a rise in accuracy from 83.79% in the original model to 85.32% in the improved model (Fig 4E). These findings validate the excellent quality of the final model.

## Disulfide modification of the vaccine construct

The Disulfide by Design 2.12 server predicted 36 potential disulfide bond pairs among the amino acid residues. Considering their bond energy value and Chi3 values, we selected three key pairs likely located in the flexible loop region of the vaccine protein (S5 Table). Cysteine was substituted for the residues (37ALA-44GLY), (113ALA-154TYR) and (232TYR-257ALA) in the refined tertiary model of the final vaccine protein. This modification enhanced the peptide's thermal stability, as illustrated in Fig 5.

**Prediction of B-cell epitopes in the engineered vaccine.** ElliPro server was used to examine the 3D structure of the designed multi-epitope vaccine construct for the purpose of identifying potential epitopes. This analysis revealed a total of 6 epitopes, including 3 linear and 3 discontinuous sequences (S6 Table). The 3D structures depicting the location of these epitopes within the final construct are presented in Fig 6

## Vaccine-TLR-4 docking

Docking simulations utilizing ClusPro were conducted to examine the interaction between the vaccine construct and TLR-4 receptors in humans and mice. The negative ΔG values obtained from the PRODIGY webserver suggest a strong binding affinity between the vaccine candidates and TLRs. Lower ΔG values indicate a high affinity for binding, and the computed energy scores are provided in Table 3. PDBsum analysis offered insights into the specific interactions between the vaccine construct's residues and the extracellular domain (ECD) of human and mouse TLR4. Table 4 summarizes the number of established salt bridges, hydrogen bonds, and non-bonded contacts with the TLR ECDs, while Fig 7 visually depicts these

## Predicted Linear Epitope(s)

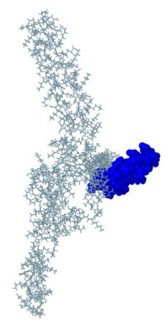

Number of residues: 39, Score: 0.752

Number of residues: 56, Score: 0.743

Number of residues: 47, Score: 0.711

## Predicted Discontinuous Epitope(s):

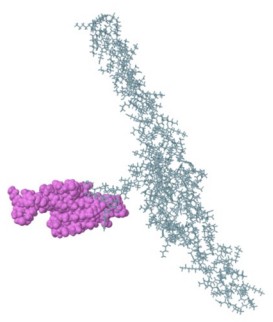
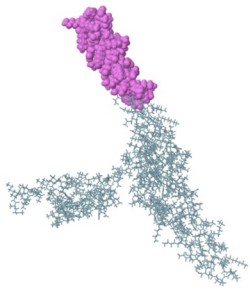
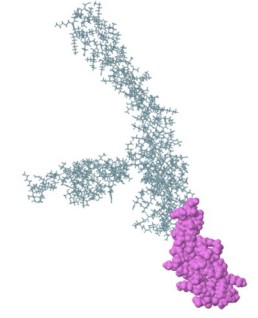

Number of residues: 60, Score: 0.764

Number of residues: 54, Score: 0.753

Number of residues: 51, Score: 0.689

**Fig 6. Predicted B-cell epitopes in the vaccine construct.** The epitopes are visualized as blue and pink spheres on the 3D structure of the protein. The number of residues in each epitope and their corresponding scores are indicated.

interactions. Notably, the human and mouse TLR4 complexes exhibited the most negative ΔG values, suggesting the strongest binding affinities.

### Normal mode analysis

To assess the conformational dynamics of the docked human and mouse TLR4-vaccine complexes, normal mode analysis was conducted on the iMODS server. For the human TLR4

**Table 3. Binding affinities and PDB IDs of TLR4 complexes.**

| TLR Name | PDB ID | Lowest Energy | ΔG value |
|----------|--------|---------------|----------|
| Human TLR4 | 4G8A | -1345.7 | -15.3 |
| Mouse TLR4 | 2Z64 | -1209.4 | -14.3 |

**Table 4. Interactions of residues among the docked complexes.**

| TLR Name | No. of Salt Bridges | No. of Hydrogen Bonds | No. of Non-Bonded Contacts |
|---|---|---|---|
| Human TLR4 | 2 | 4 | 77 |
| Mouse TLR4 | 6 | 9 | 145 |

complex, the deformability plot (Fig 8A) indicated specific regions of increased flexibility, likely corresponding to loops or linkers that are crucial for antigen presentation or receptor interaction. Similarly, the mouse TLR4 complex's deformability (Fig 8A) showed notable peaks, suggesting regions of flexibility that could play a key role in immune system engagement. The B-factor analysis (Fig 8B) for both complexes demonstrated a strong alignment between NMA-predicted and PDB-derived fluctuations, affirming the reliability of the modeled structures. The eigenvalue distribution for the human TLR4 complex (Fig 8C), with the first mode having an eigenvalue of 3.849026e-07, and for the mouse TLR4 complex (Fig 8C),

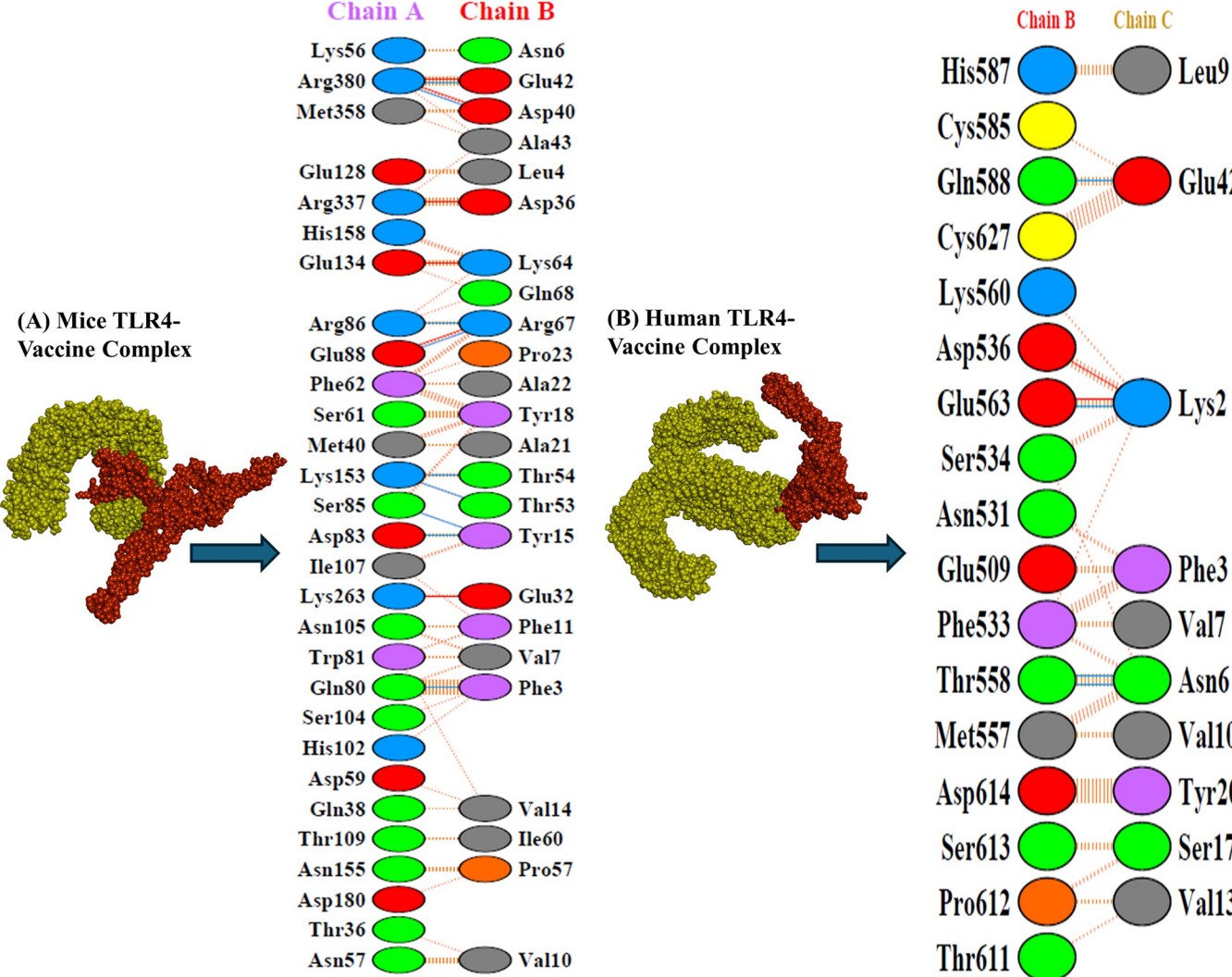

**Fig 7. Molecular interactions of the vaccine construct with TLR4.** (A) Visual representation of the interactions between the vaccine construct and mouse TLR4 (B) Visual representation of the interactions between the vaccine construct and human TLR4.

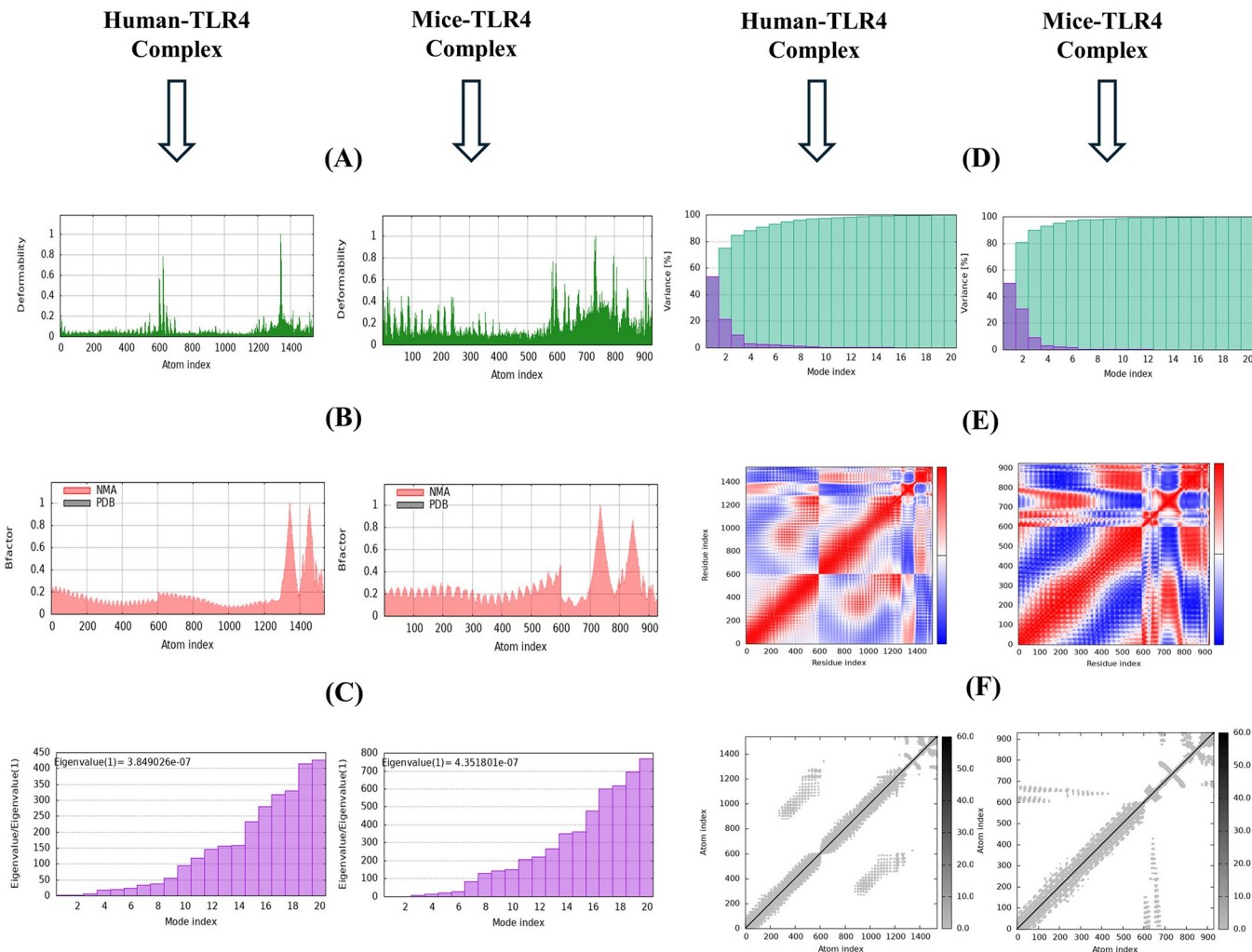

**Fig 8. Normal mode analysis of human and mouse TLR4-vaccine complexes.** Molecular dynamics simulations of the vaccine constructs with human TLR4 (left column) and mouse TLR4 (right column). (A) Deformability, (B) B-factor, (C) eigenvalues, (D) variance map, (E) covariance map, and (F) elastic network analysis. These results provide insights into the flexibility, stability, and interactions within the vaccine constructs and their complexes with TLR4.

with an eigenvalue of 4.351801e-07, suggested that both complexes possess a balanced mix of flexibility and rigidity. In both cases, the low eigenvalue of the first mode indicated flexible global motions, while subsequent modes reflected increasingly stiff, localized movements. The variance plots (Fig 8D) for both complexes showed that a significant portion of the overall motion is captured within the initial modes, emphasizing their importance in the constructs' dynamics. The covariance maps (Fig 8E) highlighted correlated and anti-correlated motions between residues in both complexes, pointing to regions of functional significance, potentially involved in receptor binding or structural adjustments. Finally, the elastic network models (Fig 8F) supported these findings, showing robust interaction networks that maintain structural integrity while allowing necessary conformational flexibility in both complexes. These analyses collectively validate the structural integrity and dynamic competence of both the human and mouse TLR4 docked with vaccine constructs, reinforcing their potential as promising candidates for further experimental validation.

## All atom molecular dynamics simulation

The molecular dynamics (MD) simulation results provide a comprehensive understanding of the dynamic behavior of the MEVC-Human TLR4 complex. The Root Mean Square Deviation (RMSD) analysis (Fig 9A) shows a gradual increase over the 100 ns simulation period, stabilizing around 60 ns with an average RMSD of 8.328 ± 2.708 Å, indicating significant conformational changes and equilibrium after approximately 60 ns. This is complemented by the Root Mean Square Fluctuation (RMSF) analysis (Fig 9B), which highlights the flexibility of individual residues, with notable peaks observed around residues 150–200 and 700–750, possibly corresponding to flexible loop regions or regions undergoing significant motion. Further insights are provided by the hydrogen bond analysis (Fig 9C), which depicts the time-dependent behavior of hydrogen bonds within the complex, with an average hydrogen bond length of 1.401 ± 0.017 Å. This suggests dynamic interactions within the binding interface throughout the simulation. Lastly, the Radius of Gyration (Rg) plot (Fig 9D) reveals minor changes in the overall compactness of the complex, with an average Rg value of 33.849 ± 0.006 Å, showing slight fluctuations before stabilization. Together, these analyses interlink to paint a detailed picture of the structural stability, conformational changes, and dynamic interactions within the MEVC-Human TLR4 complex during the simulation.

Furthermore, the MD simulation results for the MEVC-Mice TLR4 complex (Fig 10) reveal several key insights into its dynamic behavior. The Root Mean Square Deviation (RMSD) analysis (Fig 10A) shows that the complex maintains a stable conformation throughout the 100 ns simulation period, with an average RMSD of 2.519 ± 0.005 Å, suggesting minimal deviations from its initial structure. The Root Mean Square Fluctuation (RMSF) analysis (Fig 10B) highlights the flexibility of specific residues within the complex, with an average RMSF of 5.695 ± 0.043 Å. Notable peaks are observed around residues 100–150 and 500–550, indicating

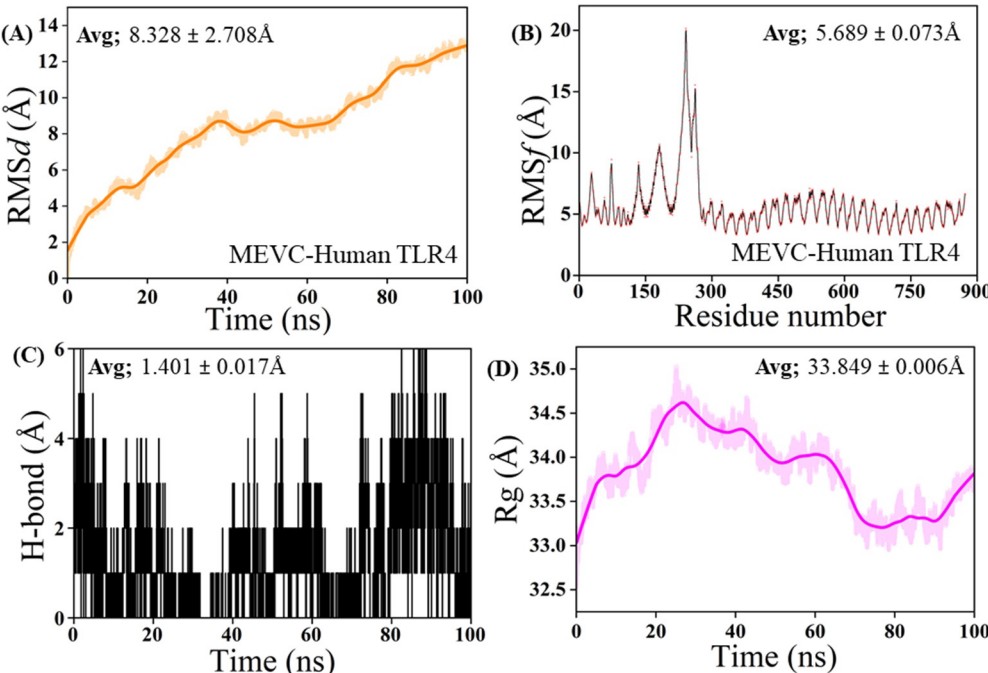

**Fig 9. Molecular dynamics simulation of MEVC-human TLR4 complex.** Molecular dynamics simulation of the MEVC-Human TLR4 complex. (A) Root-mean-square deviation (RMSD) of the protein backbone over time. (B) RMSF of individual residues. (C) Hydrogen bond formation and breakage. (D) Radius of gyration (Rg) over time.

localized flexibility in these regions. The hydrogen bond analysis (Fig 10C) indicates that the average hydrogen bond length is 2.249 ± 0.016 Å, with dynamic fluctuations over time, reflecting transient interactions within the binding interface of the complex. Lastly, the Radius of Gyration (Rg) plot (Fig 10D) shows that the complex maintains a relatively consistent shape and compactness throughout the simulation, with an average Rg value of 31.960 ± 0.005 Å. These findings collectively provide a comprehensive understanding of the structural stability, residue flexibility, and interaction dynamics within the MEVC-Mice TLR4 complex during the simulation period.

## Simulation of immune response

C-ImmSim server evaluated the potential of the designed protein to trigger an immune response. Fig 11 depicts the predicted primary and secondary immune responses within a virtual host. The analysis suggests that the vaccine candidate successfully stimulates an appropriate immune response. The initial (primary) response involves an increase in IgM levels, followed by a robust secondary response evident by a rise in IgG1, IgG2a, and IgG subclasses. This secondary response also leads to a decrease in antigen concentration (Fig 11A–11C). Furthermore, significant increases in cytokine and interleukin levels are observed after immunization (Fig 11F). The analysis also predicts a considerable expansion of T helper (Th) and cytotoxic T (Tc) cell populations, alongside the development of memory cells (Fig 11D and 11E). Additionally, the simulation suggests an increase in the production of macrophages and dendritic cells, crucial components of the immune system (Fig 11H and 11I). These findings hold promise for the development of an effective immune response against *Streptococcus pneumoniae*.

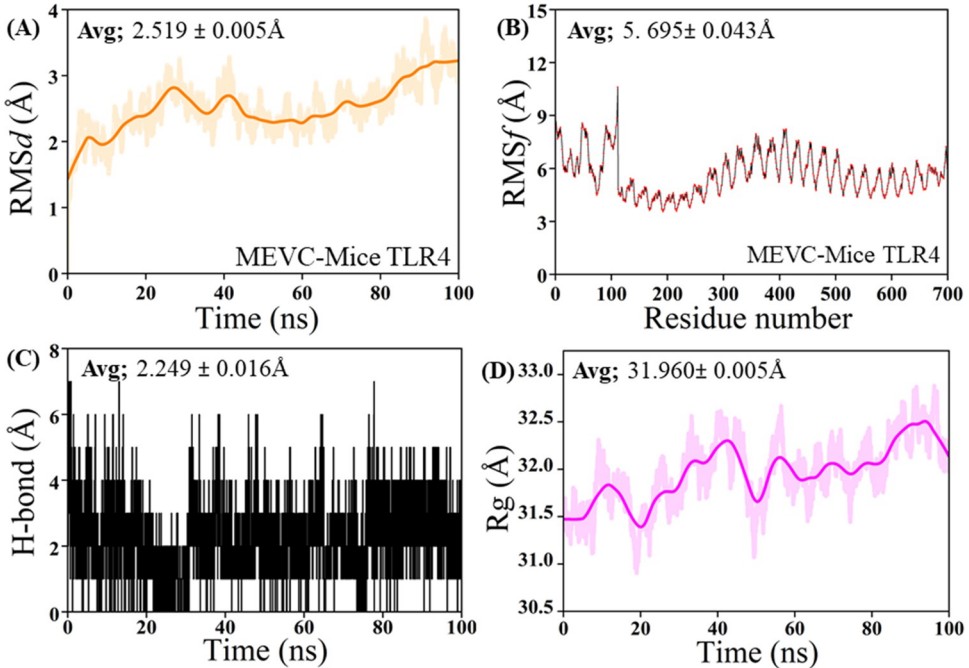

**Fig 10. Molecular dynamics simulation of MEVC-mice TLR4 complex.** Molecular dynamics simulation of the MEVC-Mice TLR4 complex. (A) Root-mean-square deviation (RMSD) of the protein backbone over time. (B) RMSF of individual residues. (C) Hydrogen bond formation and breakage. (D) Radius of gyration (Rg) over time.

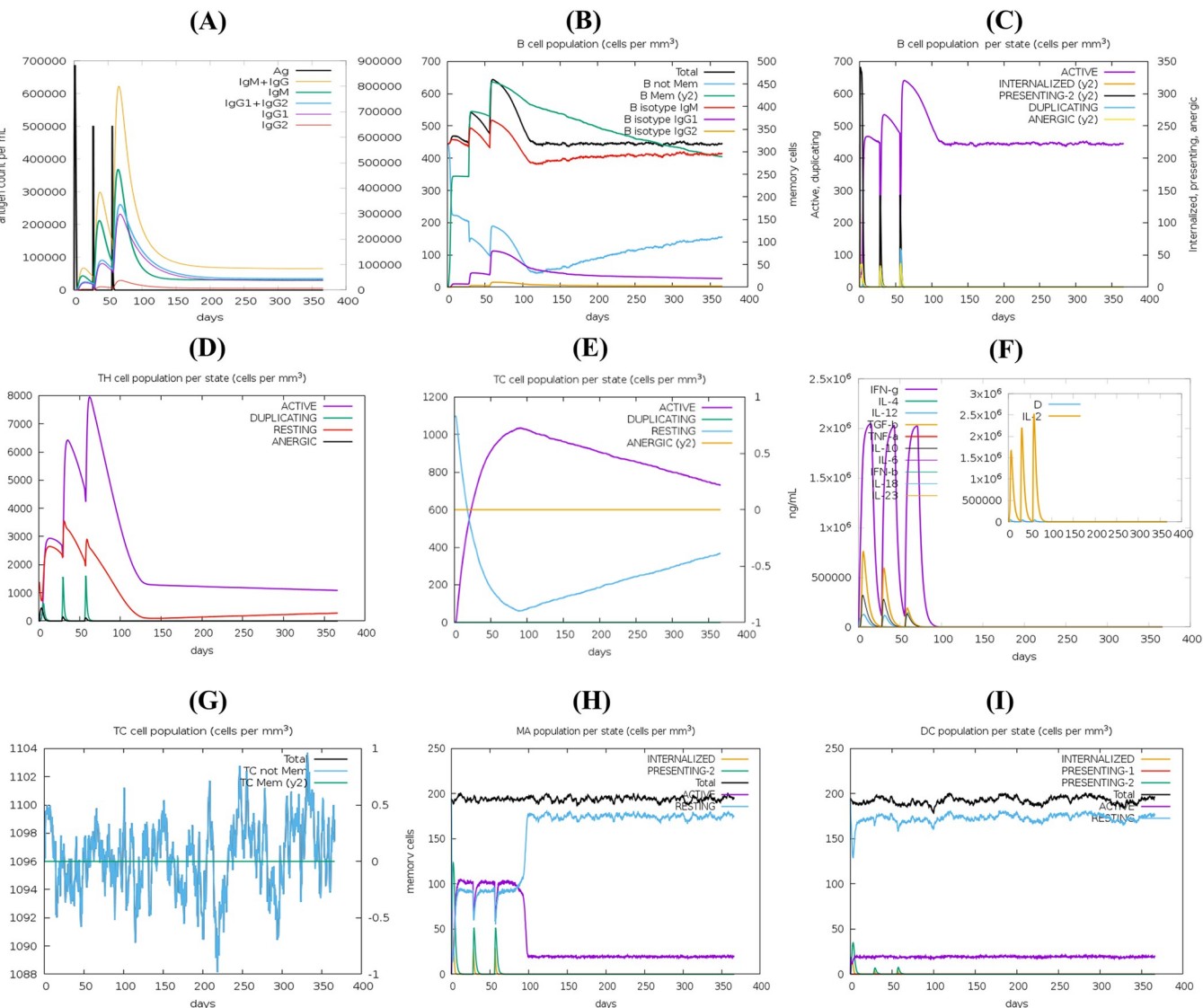

**Fig 11. Immune response simulation to vaccine antigen.** The immunological response to the vaccine antigen after three doses given at intervals of 1, 84, and 168 hours was simulated using the C-ImmSim server. The results depict the dynamics of various immune cell populations and parameters, including: (A) antigen and immunoglobulin levels, (B) B cell population, (C) B cell population distribution by state, (D) helper T cell population, (E) cytotoxic T cell population distribution by state, (F) cytokine production, (G) total cytotoxic T cell population, (H) macrophage population distribution by state, and (I) dendritic cell population distribution by state. The conducted simulations offer significant insights into the prospective effectiveness and safety of the vaccination candidate.

## Vaccine virtual expression & cloning

The optimized codon sequence shows the Codon Adaptation Index (CAI) of 0.93. which signifies a strong potential for expression in this bacterial system. This value is very close to the ideal value of 1.0, indicating optimal codon usage. Moreover, the average GC content for the vaccine construct was determined to be 59.12%, which is included in the recommended range of 30% to 70% for *E. coli* production. The Computational Fluid Dynamics (CFD) analysis of the sequence demonstrated a 0% value, indicating the absence of any atypical tandem codons in the optimized sequence. Tandem uncommon codons can diminish the effectiveness of translation or possibly disrupt the operational mechanism of translation. Finally, to facilitate

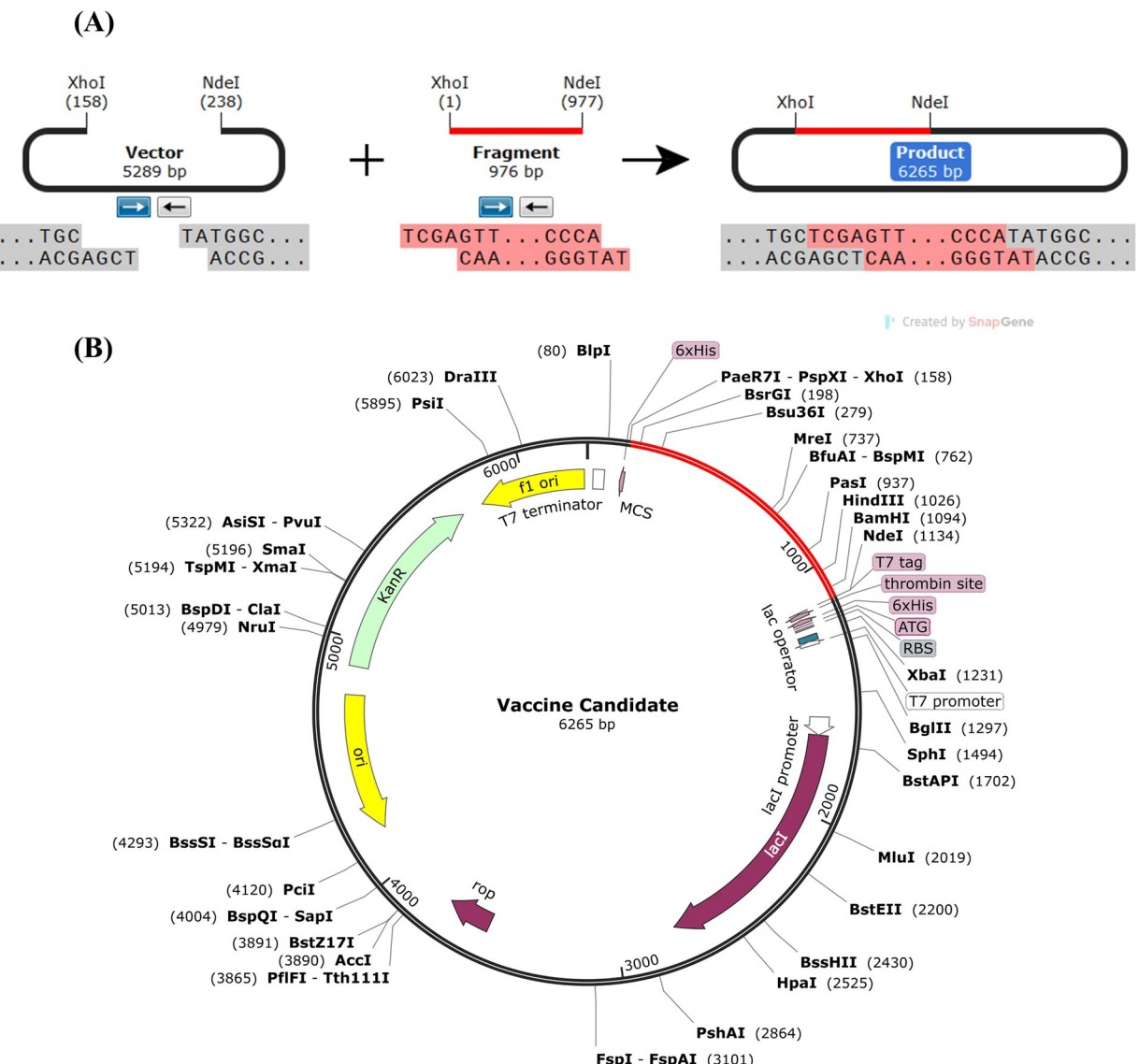

**Fig 12. Computational cloning of vaccine construct into pET28a (+) vector.** (A) A diagram illustrating the cloning process, showing the pET28a (+) vector, the vaccine construct fragment, and the final cloned product. Between the NdeI and XhoI restriction sites, the vaccine fragment is introduced. (B) Detailed map of the final construct, indicating the various restriction enzyme sites, promoters, terminators, and other elements. The vaccine construct is highlighted in red. The total length of the final construct is 6265 base pairs.

expression, the improved vaccination sequence was incorporated into the pET28a (+) plasmid vector using the SnapGene software. This insertion occurred between the NdeI and XhoI restriction sites, as depicted in Fig 12.

## Discussion

Pneumococcal infections remain a major global health concern, especially for young children and older adults [93]. These infections are caused by the bacterium *Streptococcus pneumoniae*, which has a sugary outer coat (capsular polysaccharide) that comes in various forms (sero-types). This coat allows *S. pneumoniae* to be categorized into 95 distinct serotypes, with ten particularly common ones (4, 6, 9V/9A, 14, 15F/15A, 15B/15C, 18, 19F, 19A, and 23F) being

responsible for a majority of childhood pneumococcal infections worldwide [94]. While antibiotics can fight these infections, their overuse can lead to resistant bacteria. Vaccination offers a more effective preventative approach [95]. Many countries have recently incorporated pneumococcal vaccination programs for children and older people into national immunization plans [96]. Currently available Pneumococcal conjugate vaccines (PCVs) and pneumococcal polysaccharide vaccines (PPVs) can provide protection against different types of pneumococcal disease. However, these vaccines may not be completely effective due to a phenomenon called serotype replacement [97]. Furthermore, the high cost and production challenges associated with pneumococcal conjugate vaccines (PCVs) limit their accessibility in developing countries [98]. An alternative approach involves the use of protein or peptide fragments as vaccine candidates. Several studies have investigated the potential of these proteins, both individually and in combination, for use in pre-clinical vaccine development [30]. Notably, pneumococcal proteins expressed across all serotypes hold promise for creating vaccines that are effective against multiple strains [30].

*In silico* techniques have become a compelling tool for streamlining vaccine research, significantly reducing both time and cost [99–101]. Immunoinformatics tools, for instance, offer substantial benefits in epitope identification. These tools hold applications in various aspects of epitope mapping, including designing peptide-based vaccines, enhancing our understanding of immune system function, and even predicting epitopes for disease diagnosis [102, 103]. These advancements allow researchers to identify specific protein regions that trigger B and T cell immune responses, aiding in the development of vaccines based on these epitopes. Epitope-based subunit vaccines offer numerous benefits over conventional vaccines, including enhanced safety, stability, and precision due to their targeted approach. Additionally, they are less expensive to produce, eliminating the need for extensive microbial culturing and minimizing *in vitro* experiments, ultimately leading to faster development timeframes [104, 105]. However, limitations exist, such as potentially low immunogenicity due to rapid degradation by enzymes and possible issues with immune receptor recognition. One way to address these limitations is by creating adjuvanted vaccinations that enhance the transportation of antigens to immune cells and hinder their breakdown inside the body [106]. Therefore, melittin has been incorporated into this study as an adjuvant to enhance the vaccine immunogenicity. However, future studies will be required to optimize its use and ensure its safety.

Our study began with a thorough review of existing research, concentrating on *S. pneumoniae* proteins that had been previously identified as potential vaccine candidates by other scientists. Subsequently, a subset of virulent proteins was identified based on their documented pathogenicity and capacity to stimulate immune responses. Bioinformatics analysis was employed to screen these selected proteins utilizing the Vaxijen and PSORTb web tools. Based on the highest scores generated by these tools, CbpA and PspA were chosen as target proteins for the construction of a multi-epitope vaccine. Vaxijen scores of 0.7721 and 0.6565 for CbpA and PspA, respectively, significantly exceeded the cutoff value of 0.4, strongly suggesting their antigenic potential. Furthermore, PSORTb scores of 9.73 for both proteins, well above the cutoff of 0.75, indicate a high probability of extracellular localization, a critical factor for vaccine efficacy (S1 Table).

CbpA, involved in the attachment of pneumococcus to host cells and facilitating immune system evasion, has already shown promise in previous studies as a vaccine component within a multivalent protein construct [107]. Similarly, PspA has demonstrated effectiveness in protecting animal models from fatal *S. pneumoniae* infection, and clinical trials have confirmed that recombinant PspA can elicit strong antibody responses in humans [38, 108–111]. Given the demonstrated potential of CbpA and PspA, our work aimed to construct an ideal multi-epitope vaccine by encompassing both T-cell and B-cell epitopes to generate a targeted

immune response against these antigens [112]. We utilized various bioinformatics approaches to predict T and B cell epitopes derived from CbpA and PspA proteins. In selecting the most promising candidates, we considered immunogenic properties such as antigenicity, toxicity, allergenicity, the presence of transmembrane helices, and interferon-gamma (IFN-γ) production. Based on these criteria, we prioritized B and T cell epitopes with high scores and overlap across different servers. The top-ranked peptides from CbpA and PspA were then chosen for further investigation (Fig 3).

The designed vaccine construct consists of 327 amino acids and includes the adjuvant melittin, a PADRE sequence, and various linker peptides. Melittin, a potent cationic peptide known for its immunostimulatory properties, is included to enhance the vaccine's immunogenicity by promoting antigen uptake, maturation, and presentation by antigen-presenting cells. The adjuvant is positioned at the N-terminus of the vaccine protein, followed by a PADRE sequence linked via an EAAAK linker. The PADRE (Pan-DR Epitope) sequence is incorporated to provide broad immunogenicity by binding to multiple HLA-DR alleles, ensuring a robust helper T-cell response across different genetic backgrounds. This sequence is critical for enhancing the vaccine's efficacy by facilitating the activation of CD4+ T cells, which are essential for a sustained immune response. To maintain the independent immunogenicity of B-cell epitopes (BCLs), flexible KK linkers are employed between them. These linkers allow for optimal spatial arrangement of the BCLs without interfering with their antigenicity. To ensure efficient recognition and separation of the multi-epitope regions containing helper T-lymphocyte (HTL) and cytotoxic T-lymphocyte (CTL) epitopes, GPGPG and GGGGS linkers are strategically placed within the construct. These linkers serve as spacers, promoting proper folding and accessibility of the epitopes for immune recognition.

The physicochemical characteristics of the construct were assessed utilizing the ExPasy ProtParam server, as shown in Table 5. The predicted molecular weight of 34.52 kDa falls within a favorable range for vaccine development since proteins with a mass below 110 kDa are often more straightforward to isolate and purify. This molecular weight is also beneficial for techniques like SDS-PAGE and Western blotting. The theoretical pI of 9.10 suggests potential for manipulating the net charge during isoelectric focusing, which can be useful for

**Table 5. Protein's physicochemical characteristics as determined by the ExPasy ProtParam server.**

| | |
|---|---|
| Amino acid composition (number of residues): | Ala (55), Arg (3), Asn (10), Asp (21), Gln (8), Glu (29), Gly (36), His (1), Ile (9), Leu (11), Lys (56), Met (4), Phe (4), Pro (16), Ser (13), Thr (17), Trp (3), Tyr (16), Val (15) |
| Number of total amino acids: | 327 |
| Molecular weight: | 34.52 kDa |
| Theoretical pI: | 9.10 |
| Estimated half-life (mammalian reticulocytes, *in vitro*): | 30 hours |
| Estimated half-life (yeast, *in vivo*): | >20 hours |
| Estimated half-life (*E. coli*, *in vivo*): | >10 hours |
| Number of total negatively charged residues (Asp + Glu): | 50 |
| Number of total positively charged residues (Arg + Lys): | 59 |
| Instability index: | 26.99 |
| Aliphatic index: | 53.98 |
| GRAVY (grand average of hydropathicity): | -0.905 |

optimizing protein purification by ion exchange chromatography. Our analysis predicted a high level of stability for the vaccine candidate, with an instability index of 26.99. This value is much lower compared to the findings of previous research on vaccine development by Bahadori et al. (2022) [2], Shafaghi et al. (2023) [113], Dorosti et al. (2019) [21], and Mazumder et al. (2023) [114]. Generally, proteins with an instability index lower than 40 are regarded as stable. This suggests our vaccine design might be less prone to degradation, potentially leading to improved effectiveness and longevity compared to previous findings. The aliphatic index of 54.98 further suggests the thermostability of the designed protein. The estimated half-lives of the construct in mammalian reticulocytes, yeast, and *E. coli* are all greater than 10 hours. This implies that the protein exhibits a high degree of stability and has an extended duration of existence within living organisms. A GRAVY score of -0.905 indicates that the vaccine is hydrophilic, implying a high attraction to water molecules and a high probability of substantial solubility. Strong antigen-water interactions are often linked to enhanced interaction with immune system components, particularly B cell receptors. Upon overexpression in *E. coli*, solubility studies also indicated elevated levels of solubility. The vaccine's antigenicity was assessed as 1.37 and 0.89 using VaxiJen and ANTIGENpro servers, respectively. These results demonstrate a substantial improvement compared to the findings of Bahadori et al. (2022) [2], Shafaghi et al. (2023) [113], Dorosti et al. (2019) [21] and Mazumder et al. (2023) [114], Munia et al. (2021) [115].

In order to enhance the precision of the 3D model for the vaccine candidate, the initial structure produced by the Robetta server underwent further refinement using the GalaxyRefine webserver. The accuracy of both the original and improved models was evaluated by utilizing various methods, including Ramachandran plots, ProSA Z-scores, Errat scores, and Verify3D scores. The results showed a notable improvement in the quality of the vaccine's predicted tertiary structure after refinement (Table 6). The Ramachandran plot analysis conducted after refinement revealed that the majority of amino acid residues (93.4%) were located in favorable areas, indicating a model of high quality. In addition, the improved model produced a ProSA Z-score of -5.12, an Errat score of 97.46, and a Verify3D score of 85.32%, further supporting its potential as a promising multi-epitope vaccine candidate.

Following the finalization and quality assessment of the vaccine model, the analysis focused on identifying potential targets for the immune system. ElliPro, a computer program, scanned the 3D model and identified both linear (continuous) and discontinuous (non-linear) epitopes. These epitopes received high scores (0.689–0.764), demonstrating their ability to interact with B cells. This implies that the vaccine can effectively activate the humoral immune response, which is crucial for defending against pneumococcal infections. Toll-like receptors (TLRs) have a crucial function in identifying pathogens and initiating the innate immune response to microbial infections [116]. To evaluate the vaccine's ability to stimulate the immune system, we employed docking simulations to investigate how it might bind to TLR4 receptors in both mice and humans. The results indicated strong binding affinity between the vaccine construct and TLR4 from each species, suggesting that the vaccination has the capacity

**Table 6. Vaccine evaluation results before and after refinement.**

| Server | Before refinement | After refinement |
|---|---|---|
| ERRAT | 96.11 | 97.46 |
| Verify 3D | 83.79 | 85.32 |
| PROCHECK | 91.9% of residues were in the most favored zone, 5.9% were in the allowed region, and 1.1% were in the disallowed region. | 93.4% of residues were in the most favored zone, 5.5% were in the allowed region, and 0.4% were in the disallowed region. |
| ProSA | – 5.24 | – 5.12 |

to stimulate these receptors and elicit an immunological response against *S. pneumoniae*. Notably, both constructs exhibited stronger binding affinity to TLR4, reflected by the most favorable energy scores (lowest values of -1345.7 and -1209.4 for human and mouse TLR4, respectively). Here, lower energy scores indicate a tighter binding complex between the receptor and the vaccine, which is crucial for stimulating a robust immunological response. Our docking score is lower in comparison to the *in silico* studies conducted by Bahadori et al. (2022) [2], Shafaghi et al. (2023) [113], and Mazumder et al. (2023) [114]. Additionally, the negative ΔG values (-15.3 and -14.3) suggest stable binding. For a deeper understanding of how the vaccine interacts with TLRs, we analyzed the intermolecular protein-protein interactions using PDBsum. This analysis identified the establishment of hydrogen bonds between the vaccine and TLR4. Specifically, 6 hydrogen bonds were observed between the vaccine and human TLR4, while 9 were found between the vaccine and mouse TLR4. Both complexes also displayed salt bridge interactions, which are the strongest type of non-covalent interaction and contribute significantly to biomolecular stability. We performed normal mode analysis using iMODS to predict the complex's stability. The analysis indicated a low deformability of the complexes, with high eigenvalues (3.849026e-07 and 4.351801e-07), suggesting that minimal energy is required to deform the structures of the vaccine-TLR4 complexes. The deformability graphs (Fig 8A) further supported the stability of the complexes, showing minimal hinge regions and a low degree of deformation for individual amino acid residues. Analysis of correlated amino acids and stiffer regions also yielded positive results for both vaccine-TLR4 complexes. Furthermore, molecular dynamics simulations using the Amber v.22 tool provided additional insights into the stability of complexes. Key parameters such as root mean square deviation (RMSD), root mean square fluctuation (RMSF), radius of gyration (Rg), and hydrogen bonding patterns were analyzed. The results indicated that the complexes maintained stable conformations over time, with consistent hydrogen bonding patterns and minimal fluctuations in RMSD and Rg, further confirming the robustness of the vaccine-TLR4 interactions. The immune simulation mirrored a typical immune response. Repeated antigen exposure led to a gradual increase in overall immune activity. Notably, the simulation predicted the formation of memory B and T cells, with memory B cells having the capacity to persist for several months. Interestingly, there was a significant increase in the levels of interferon-gamma (IFN-γ) and interleukin-2 (IL-2) following the initial injection, and these levels remained elevated with subsequent exposures. This suggests a potential increase in T helper (Th) cell populations, which could lead to robust antibody (Ig) production and a strong humoral immune response. The simulation also indicated promising activity of dendritic cells and macrophages, essential components of the adaptive immune system. Additionally, the simulation predicted the activation of components within the innate immune system. The simulation also indicated satisfactory activity of both dendritic cells and macrophages, key players in the immune system. The Simpson Diversity Index (D) suggests potential variations in immune responses, warranting further investigation into clonal specificity. For optimal production of proteins in *E. coli* bacteria (K12 strain), the designed vaccine construct was optimized for codon usage using the GenSmart Codon Optimization Tool server. A well-established principle is that a gene's GC content and codon adaptation index (CAI) influence the level of protein expression in the host organism [90]. In this study, we optimized the vaccine sequence by ensuring a GC content of 59.12% and a CAI value of 0.93. These values fall within a well-defined range for achieving high expression levels of the multi-epitope protein in *E. coli*. To mimic the cloning process, we then proceeded to insert the optimized vaccine sequence into a widely used expression vector, pET28a (+). The results from the computational analysis suggest that this vaccine candidate has promising potential against *S. pneumoniae*. However, further laboratory experiments (*in vitro* and *in vivo*) are necessary to validate these findings.

## Conclusion

The objective of this study was to develop a multi-epitope vaccine that might efficiently provide protection against *S. pneumoniae* by stimulating antibody production. The vaccine incorporates carefully selected epitopes from Choline-binding protein A (CbpA) and Pneumococcal surface protein A (PspA), identified through rigorous *in silico* screening for allergenicity and toxicity. To improve the vaccine's effectiveness, melittin was added as an adjuvant. The designed vaccine construct exhibited promising physicochemical properties, indicating stability, thermostability, and good water solubility. Docking and dynamic simulations predicted a high affinity and binding potential of the subunit vaccine protein with immune receptors, further supporting its stability over time. The immune simulation results mirrored real-world immune responses, with the potential to create a lasting immune memory against *S. pneumoniae* infections. Additionally, the vaccine sequence was optimized for efficient expression in *E. coli* and virtually cloned into the pET28a(+) vector. While *in silico* analysis yielded strong evidence for immunogenicity and stability, further laboratory experiments are necessary to confirm the effectiveness of this multi-epitope vaccine candidate.

## Supporting information

**S1 Table. Candidate vaccine proteins following bioinformatic screening for antigenicity and extracellular localization.**
(DOCX)

**S2 Table. Prediction of all B-cell epitopes from CbpA & PspA.**
(DOCX)

**S3 Table. Prediction of all CTL epitopes from CbpA & PspA.**
(DOCX)

**S4 Table. Prediction of all HTL epitopes from CbpA & PspA.**
(DOCX)

**S5 Table. Predicted disulfide bond partners by disulfide by design 2.0.**
(DOCX)

**S6 Table. Predicted discontinuous epitope(s) for the construct.**
(DOCX)

## Author Contributions

**Conceptualization:** Shahina Akter.

**Data curation:** Md. Nahian.

**Formal analysis:** Md. Nahian, Muhammad Shahab, Md. Rasel Khan, Shopnil Akash, Jonas Ivan Nobre Oliveira.

**Investigation:** Sanjana Fatema Chowdhury, Mohammad Ariful Islam, Shamima Begum, Ahashan Habib.

**Methodology:** Md. Nahian, Muhammad Shahab, Md. Rasel Khan.

**Project administration:** Shahina Akter.

**Resources:** Tanjina Akhtar Banu, Barna Goswami, Aftab Ali Shaikh, Jonas Ivan Nobre Oliveira.

**Software:** Jonas Ivan Nobre Oliveira.

**Supervision:** Ahmed Abu Rus'd, Shahina Akter.

**Validation:** Mohammad Ariful Islam, Shamima Begum, Ahashan Habib, Aftab Ali Shaikh, Jonas Ivan Nobre Oliveira, Shahina Akter.

**Visualization:** Md. Rasel Khan, Shopnil Akash, Tanjina Akhtar Banu, Murshed Hasan Sarkar, Barna Goswami, Sanjana Fatema Chowdhury.

**Writing – original draft:** Md. Nahian, Shopnil Akash.

**Writing – review & editing:** Shahina Akter.

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
