## [Decision Letter · Decision Letter 0]

17 Jul 2024

PONE-D-24-24274Development of a Broad-Spectrum Epitope-Based Vaccine Against Streptococcus pneumoniaePLOS ONE

Dear Dr. Akter,

Thank you for submitting your manuscript to PLOS ONE. After careful consideration, we feel that it has merit but does not fully meet PLOS ONE’s publication criteria as it currently stands. Therefore, we invite you to submit a revised version of the manuscript that addresses the points raised during the review process.

We look forward to receiving your revised manuscript.

Kind regards,

Rajesh Kumar Pathak, Ph.D.

Academic Editor

PLOS ONE

Journal Requirements:

Additional Editor Comments:

The manuscript in its present form lacks clarity and requires extensive revision. Information about the half-life of the vaccine needs to be mentioned. The results of the molecular dynamics simulation need to be explained in detail during revision. Important analyses such as RMSD and RMSF are missing and should be included. Additionally, the concerns raised by the reviewers need to be thoroughly addressed in the revised manuscript.

Reviewers' comments:

Reviewer's Responses to Questions

**Comments to the Author**

1. Is the manuscript technically sound, and do the data support the conclusions?

Reviewer #1: Yes

Reviewer #2: Partly

2. Has the statistical analysis been performed appropriately and rigorously? 

Reviewer #1: N/A

Reviewer #2: No

3. Have the authors made all data underlying the findings in their manuscript fully available?

Reviewer #1: Yes

Reviewer #2: Yes

4. Is the manuscript presented in an intelligible fashion and written in standard English?

Reviewer #1: Yes

Reviewer #2: Yes

5. Review Comments to the Author

Reviewer #1: 1. Use Escherichia coli (E. coli) for the first time in the manuscript, and E. coli will be used from then on. Please do the same for toll-like receptors (TLRs).

2. The words "in silico", "in vivo", and "invitro" should be written in italics.

3. In the topic of this work, the word "prediction" is not the right word, use "prediction" instead.

4. Modeling, refining, validation of the three-dimensional structure 3D structure of CbpA and PspA proteins, as well as predicting structural B-cell epitopes based on their 3D structure, are trivial tasks. Please remove these parts from the manuscript.

5. Authors must provide a rationale for using each linker in the vaccine structure.

6. The authors only reported the value of the quality assessment parameters of the refined 3D model if they did not mention the reason for choosing the model. The output of the GalaxyRefine server is 5 refined models, in which the quality evaluation parameters of the refined 3D model are presented. Please provide this table in the manuscript, describe each parameter in detail, and explain why this model was chosen.

7. Authors should provide the definition of each physicochemical parameter in the Materials and Methods section.

8. The authors reported the results of the molecular dynamics simulation in a very general way. Please explain the results of this section in detail.

9. The authors should perform structural disulfide engineering of the 3D structure of the vaccines

10. Please perform the molecular docking between T-cell epitopes and MHC molecules.

11. Please compare the characteristics of the vaccines designed in this study with the vaccines designed in other authors' studies. For this, please use the following references and cite them.

https://doi.org/10.3390/vaccines11020263

https://doi.org/10.2174/1573409919666230612125440

https://doi.org/10.1080/07391102.2023.2258403

https://doi.org/10.1016/j.ijbiomac.2024.131517

https://doi.org/10.1007/s12033-023-00949-y

Reviewer #2: In the current study, authors constructed a multi-epitope vaccine using bioinformatics analysis. Here are some comments needs to be addressed.

Minor

Keep the font size constant throughout the paper. See 2.6 is smaller compared to others.

Major

1. Authors identified 23 virulence protein in S. pneumoniae based on their search. I am curious to know in general how many total proteins are encoded in the S. pneumoniae genome. What will be the scores of other proteins not covered in this study. Current score of top2 proteins is not very high. Also, has somebody previously reported about these proteins?

2. I am not very confident with the various webserver-based results as the values are too low for many of the analysis ranging from 0.4-0.6 in many cases. I am concerned regarding the efficacy of the vaccine.

3. Authors have not mentioned about the stability of half-life of the vaccine in the blood?

6. PLOS authors have the option to publish the peer review history of their article (what does this mean?). If published, this will include your full peer review and any attached files.

Reviewer #1: No

Reviewer #2: **Yes: **Piyush Agrawal, Ph.D.

---

## [Author Response · Author response to Decision Letter 0]

17 Oct 2024

PONE-D-24-24274

Development of a Broad-Spectrum Epitope-Based Vaccine Against Streptococcus pneumoniae

PLOS ONE

Comments from the Editors:

The manuscript in its present form lacks clarity and requires extensive revision. Information about the half-life of the vaccine needs to be mentioned. The results of the molecular dynamics simulation need to be explained in detail during revision. Important analyses such as RMSD and RMSF are missing and should be included. Additionally, the concerns raised by the reviewers need to be thoroughly addressed in the revised manuscript.

Ans: We thank the academic editor for the careful reading of the manuscript and the constructive remarks. We have addressed your comments to improve and clarify the manuscript throughout. As suggested, we have incorporated detailed information regarding the half-life of the vaccine. Initially, the vaccine construct demonstrated an unsatisfactory half-life, prompting us to optimize the adjuvant sequence. This optimization resulted in a significant improvement in the half-life, ensuring greater stability and efficacy of the vaccine. These findings are now clearly presented in the revised manuscript. To further strengthen the results, we have conducted additional molecular dynamics simulations and included crucial analyses such as RMSD, RMSF, Rg, and hydrogen bonding patterns. To facilitate the review process, we have also submitted a track-changes document highlighting the modifications made. Figures have been uploaded separately in accordance with the journal's requirements. Additionally, we have provided a point-by-point response to all reviewers' comments. (reviewers’ comments in black, our replies in blue).

Reviewer #1: 

1. Use Escherichia coli (E. coli) for the first time in the manuscript, and E. coli will be used from then on. Please do the same for toll-like receptors (TLRs).

Ans: Thank you for your valuable feedback. We have revised the manuscript to use the full term "Escherichia coli" the first time it is mentioned, with "E. coli" used in subsequent references. Similarly, we have spelled out "toll-like receptors" at their initial mention and used "TLRs" thereafter.

2. The words "in silico", "in vivo", and "invitro" should be written in italics.

Ans: Thank you for suggesting this specific point. We have made the terms "in silico," "in vivo," and "in vitro" italic throughout the manuscript as per your suggestion.

3. In the topic of this work, the word "prediction" is not the right word, use "prediction" instead.

Ans: Thank you for your comment. We have reviewed the manuscript and corrected the use of the word 'prediction' as suggested.

4. Modeling, refining, validation of the three-dimensional structure 3D structure of CbpA and PspA proteins, as well as predicting structural B-cell epitopes based on their 3D structure, are trivial tasks. Please remove these parts from the manuscript.

Ans: We appreciate your insightful comment. We have removed the sections pertaining to the modeling, refining, validation of the 3D structures of CbpA and PspA proteins, as well as the prediction of structural B-cell epitopes.

5. Authors must provide a rationale for using each linker in the vaccine structure.

Ans: We have carefully considered the rationale for each linker used in the vaccine structure and have provided a detailed explanation for their inclusion is provided in the 'Methodology' section of the revised manuscript.

6. The authors only reported the value of the quality assessment parameters of the refined 3D model if they did not mention the reason for choosing the model. The output of the GalaxyRefine server is 5 refined models, in which the quality evaluation parameters of the refined 3D model are presented. Please provide this table in the manuscript, describe each parameter in detail, and explain why this model was chosen.

Ans: Thank you for your valuable feedback. A table presenting the quality assessment parameters of all five refined 3D models generated by the GalaxyRefine server has been included in the manuscript. Each parameter has been described in detail, and a clear rationale for the selection of the final model is provided.

7. Authors should provide the definition of each physicochemical parameter in the Materials and Methods section.

Ans: Thank you for this insightful comment. The definitions of all physicochemical parameters used in the study have been clearly provided in the 'Methodology' section of the revised manuscript.

8. The authors reported the results of the molecular dynamics simulation in a very general way. Please explain the results of this section in detail.

Ans: As suggested by the reviewer, we have now provided a detailed explanation of the molecular dynamics simulation results in the revised manuscript. To gain deeper insights into the molecular dynamics, we employed the Amber v.22 tool to analyze key parameters such as RMSD, RMSF, Rg, and hydrogen bonding patterns.

9. The authors should perform structural disulfide engineering of the 3D structure of the vaccines

Ans: As suggested by the reviewer, we have now performed structural disulfide engineering on the 3D structure of the vaccines. The results of this analysis are included in the revised manuscript.

10. Please perform the molecular docking between T-cell epitopes and MHC molecules.

Ans: As suggested by the reviewer, we have now performed molecular docking between the T-cell epitopes and MHC molecules. The results of this analysis are included in the revised manuscript.

11. Please compare the characteristics of the vaccines designed in this study with the vaccines designed in other authors' studies. For this, please use the following references and cite them.

https://doi.org/10.3390/vaccines11020263

https://doi.org/10.2174/1573409919666230612125440

https://doi.org/10.1080/07391102.2023.2258403

https://doi.org/10.1016/j.ijbiomac.2024.131517

https://doi.org/10.1007/s12033-023-00949-y

Ans: Thank you for your suggestion. We have now compared the characteristics of the vaccines designed in our study with those from the studies referenced and have included the appropriate citations.

Reviewer #2: In the current study, authors constructed a multi-epitope vaccine using bioinformatics analysis. Here are some comments needs to be addressed.

Minor

Keep the font size constant throughout the paper. See 2.6 is smaller compared to others.

Ans: Thank you for your comment. We have carefully reviewed the manuscript and ensured that the font size is consistent throughout the entire document,

Major

1. Authors identified 23 virulence protein in S. pneumoniae based on their search. I am curious to know in general how many total proteins are encoded in the S. pneumoniae genome. What will be the scores of other proteins not covered in this study. Current score of top2 proteins is not very high. Also, has somebody previously reported about these proteins?

Ans: We thank you for the insightful comments. We have modified our ‘’Methodology’ section to remove any ambiguity and have also provided the following information regarding our selection criteria for the protein candidates.

Total Proteins in S. pneumoniae Genome:

The exact number of proteins encoded in the S. pneumoniae genome can vary slightly depending on the specific strain. However, recent estimates suggest that S. pneumoniae has approximately 2,000-2,300 protein-coding genes.

Potential Scores of Other Proteins:

It is possible that other proteins not included in this study could also score well in bioinformatic predictions. However, our study did not aim to conduct a comprehensive proteome analysis. Instead, we focused on proteins that have already been validated as potential vaccine candidates through various experimental studies. Numerous previous studies have identified certain proteins in S. pneumoniae that demonstrate significant efficacy as vaccine targets. Therefore, we initially selected our protein vaccine candidates based on their established immunogenic properties identified through literature mining.

Current Scores of Top 2 Proteins:

To further refine our selection, we employed VaxiJen and PsortB to predict antigenicity and subcellular localization, respectively. While we acknowledge that other proteins within the S. pneumoniae proteome may exhibit similar or even higher scores, our approach allowed us to identify two promising candidates, CbpA and PspA, with exceptional characteristics. Their VaxiJen scores of 0.7721 and 0.6565, significantly surpassing the cutoff of 0.4, strongly indicate their antigenic potential. Moreover, their PsortB scores of 9.73 each, well above the cutoff of 7.5, suggest a high probability of extracellular localization, a crucial factor for vaccine efficacy.

Previous Reports on the Selected Proteins: 

CbpA and PspA have been previously identified in the literature as potential vaccine candidates due to their roles as virulence factors in S. pneumoniae and their capacity to elicit an immune response. We also evaluated these proteins using bioinformatic tools to assess their antigenicity and subcellular localization, thereby strengthening the case for their inclusion in a multi-epitope vaccine construct. 

We are also aware of previous studies that have incorporated S. pneumoniae proteins into multi-epitope vaccine construction. However, to the best of our knowledge, the specific combination of CbpA and PspA as potential targets in a multi-epitope-based vaccine has not been previously reported. The vaccine construct we developed, consisting of epitopes from both CbpA and PspA, represents a novel approach.

2. I am not very confident with the various webserver-based results as the values are too low for many of the analysis ranging from 0.4-0.6 in many cases. I am concerned regarding the efficacy of the vaccine.

Ans: Thank you for your valuable comment. We appreciate your concern regarding the efficacy of the vaccine. With due respect, we would like to inform you that the antigenicity scores generated by VaxiJen (1.371) and ANTIGENpro (0.89) are within acceptable ranges and indicative of potential immunogenicity. Experiments conducted by other tools have also yielded promising results, suggesting the potential efficacy of our vaccine construct, which is discussed in detail in the 'Discussion' section. These results collectively support the potential of our vaccine and suggest it is suitable for subsequent wet lab experiments for further verification.

3. Authors have not mentioned about the stability of half-life of the vaccine in the blood.

Ans: Thank you for your careful remark. We have now incorporated information regarding the stability and half-life of the vaccine in the blood.

---

## [Decision Letter · Decision Letter 1]

12 Nov 2024

PONE-D-24-24274R1Development of a Broad-Spectrum Epitope-Based Vaccine Against Streptococcus pneumoniaePLOS ONE

Dear Dr. Akter,

Thank you for submitting your manuscript to PLOS ONE. After careful consideration, we feel that it has merit but does not fully meet PLOS ONE’s publication criteria as it currently stands. Therefore, we invite you to submit a revised version of the manuscript that addresses the points raised during the review process.

We look forward to receiving your revised manuscript.

Kind regards,

Rajesh Kumar Pathak, Ph.D.

Academic Editor

PLOS ONE

Journal Requirements:

**Additional Editor Comments:**

It appears that Figures 9A to 9D and Figure 10, related to the MD simulation analysis, are referenced in the manuscript text but are not included. Kindly ensure that these figures are added.

Reviewers' comments:

Reviewer's Responses to Questions

**Comments to the Author**

1. If the authors have adequately addressed your comments raised in a previous round of review and you feel that this manuscript is now acceptable for publication, you may indicate that here to bypass the “Comments to the Author” section, enter your conflict of interest statement in the “Confidential to Editor” section, and submit your "Accept" recommendation.

Reviewer #1: All comments have been addressed

Reviewer #2: All comments have been addressed

2. Is the manuscript technically sound, and do the data support the conclusions?

Reviewer #1: Yes

Reviewer #2: Partly

3. Has the statistical analysis been performed appropriately and rigorously? 

Reviewer #1: N/A

Reviewer #2: N/A

4. Have the authors made all data underlying the findings in their manuscript fully available?

Reviewer #1: Yes

Reviewer #2: Yes

5. Is the manuscript presented in an intelligible fashion and written in standard English?

Reviewer #1: Yes

Reviewer #2: Yes

6. Review Comments to the Author

Reviewer #1: The authors considered all the comments correctly. I hope this article is an important reference in the field of vaccine design

Reviewer #2: (No Response)

7. PLOS authors have the option to publish the peer review history of their article (what does this mean?). If published, this will include your full peer review and any attached files.

Reviewer #1: No

Reviewer #2: **Yes: **Piyush Agrawal

---

## [Author Response · Author response to Decision Letter 1]

17 Dec 2024

We have carefully checked the reviewers comments and revised accordingly

---

## [Editor Report · Decision Letter 2]

23 Dec 2024

Development of a Broad-Spectrum Epitope-Based Vaccine Against Streptococcus pneumoniae

PONE-D-24-24274R2

Dear Dr. Akter,

We’re pleased to inform you that your manuscript has been judged scientifically suitable for publication and will be formally accepted for publication once it meets all outstanding technical requirements.

Kind regards,

Rajesh Kumar Pathak, Ph.D.

Academic Editor

PLOS ONE

Additional Editor Comments (optional):

The manuscript can be accepted for publication.
---

## [Editor Report · Acceptance letter]

7 Jan 2025

PONE-D-24-24274R2 

PLOS ONE

Dear Dr. Akter, 

I'm pleased to inform you that your manuscript has been deemed suitable for publication in PLOS ONE. Congratulations! Your manuscript is now being handed over to our production team.

Kind regards, 

on behalf of

Dr. Rajesh Kumar Pathak 

Academic Editor

PLOS ONE